# RAISE THE BAR: ENSEMBLE-BASED ONLINE REINFORCEMENT LEARNING FOR DYNAMIC WORKFLOW SCHEDULING

## ABSTRACT

Dynamic workflow scheduling (DWS) in cloud computing poses major challenges due to unpredictable workflow arrivals, heterogeneous resources, and evolving system states. While reinforcement learning (RL) has shown promise for learning adaptive scheduling policies, existing single-policy approaches often struggle in online settings with non-stationary workloads. We propose **RAISE** (*Robust Actor-Critic Integration for Scheduling Ensembles*), an ensemble-based online RL approach designed to improve adaptability and stability in dynamic environments. RAISE maintains a set of pre-trained actors and critics that are continually updated during deployment to support robust scheduling. It integrates three key components: (1) *Value-Ranked Action Aggregation*, which combines majority voting with critic-guided tie-breaking for stable action selection; (2) *Dual Critic Ensembles with Decoupled Updates*, which balance fast adaptation and stable value estimates; and (3) *Decision-Aligned Policy Updates*, which enhance sample efficiency by updating only the actors responsible for chosen actions. Experiments on large-scale DWS benchmarks show that RAISE consistently outperforms state-of-the-art baselines in both performance and robustness, demonstrating the effectiveness of ensemble-based online RL for real-time scheduling under non-stationary conditions.

## 1 INTRODUCTION

Dynamic workflow scheduling (DWS) lies at the heart of cloud computing operations, where a continuous stream of workflows, each structured as a *directed acyclic graph* (DAG) of interdependent tasks, must be efficiently assigned for execution on a heterogeneous pool of machines in real time (Chen et al., 2025b; Shen et al., 2025; Yang et al., 2025). Each task assignment must respect precedence constraints and resource heterogeneity, while optimizing long-term performance metrics such as *average flowtime* or *total makespan*. What makes DWS especially challenging is its dynamic and uncertain nature. Workflows arrive unpredictably and machine availability fluctuates over time. The *scheduling agent* must make fast, context-sensitive decisions based solely on the current system state. It must also rapidly adapt to a live, continuously changing environment through a single, non-repeating stream of interactions.

We formulate DWS as an *online sequential decision problem*, where the scheduling agent must assign each focused task to a specific machine according to its current policy at every decision step. In practice, these assignments are typically executed immediately, making each decision effectively final. The complexity of real-world DWS arises from task dependencies, resource heterogeneity, and shifting workload dynamics. These factors make traditional heuristics such as *priority dispatching rules* (PDRs) (Topcuoglu et al., 2002; Pham & Fahringer, 2020) increasingly inadequate. Hand-crafted PDRs are *static*, unable to adapt to changing conditions, and do not learn from prior experience. Recent research has explored *deep reinforcement learning* (DRL) as a way to develop adaptive, data-driven scheduling policies that can generalize across diverse workflow patterns (Jayanetti et al., 2024; Zhu et al., 2024). However, most existing approaches focus on training policies from scratch, while often neglecting subsequent **online adaptation** to evolving environments. In contrast, practical deployment demands *robust online adaptation*, since learning from scratch is infeasible due to streaming, non-repetitive workloads and the need for consistent real-time performance.

**Online reinforcement learning** (Online RL), which continually improves a pre-trained policy using streaming data from the live environment (Hamadanian et al., 2025; Weisz et al., 2023; Zhou et al., 2025), offers a promising foundation for adaptive scheduling. However, realizing this potential is non-trivial. Existing single-policy methods, such as GOODRL (Yang et al., 2025), lack the safety mechanisms required for long-term online operations, where a single aggressive decision on a bottleneck task can cascade into system-wide delays. Applying online RL to combinatorial optimization problems such as online DWS introduces *three core challenges*. **First**, a single pre-trained policy can be unreliable in *non-stationary* environments. Its performance may deteriorate under distributional shifts or become biased by recent workload patterns, resulting in unstable or suboptimal scheduling performance (Chen & Huang, 2023; Gao et al., 2024; Jiang et al., 2023). **Second**, maintaining accurate value estimation during continual adaptation is difficult. Aggressive updates of value functions can lead to instability and loss of useful offline knowledge (Ball et al., 2023; Zhang et al., 2024b), while overly conservative updates slow down adaptation to new workload dynamics (Chen et al., 2025a; Fang et al., 2025). **Third**, the scheduling agent must learn and adapt *efficiently*, *responding in real time* to changing system states without compromising long-term performance or consistency.

In this paper, we develop **RAISE** (**R**obust **A**ctor-Critic **I**ntegration for **S**cheduling **E**nsembles), an ensemble-based online RL approach for dynamic workflow scheduling. RAISE maintains a diverse set of pre-trained actors and critics, enabling robust decision-making and continuous adaptation under non-stationary conditions. It directly addresses key challenges in *online* scheduling: resolving conflicting actor decisions, stabilizing value estimation during online policy updates, and maintaining ensemble diversity through an efficient learning process. **Our contributions are threefold:** (i) We propose *Value-Ranked Action Aggregation*, a robust decision mechanism that combines majority voting among actors with critic-guided tie-breaking based on relative value rankings. (ii) We design *Dual Critic Ensembles with Decoupled Updates*, which use a set of *adaptive critics* for fast learning and a set of *conservative critics* to preserve stable value estimates from prior knowledge. (iii) We introduce *Decision-Aligned Policy Updates*, a selective training strategy that updates only those actors responsible for chosen actions, improving sample efficiency and preserving policy diversity.

This paper identifies Online Optimization for NCO as an underexplored frontier and presents ensemble-based online RL as a principled solution, paving the way for future progress in this domain. Extensive experiments on large-scale online scenarios show that RAISE consistently outperforms state-of-the-art baselines. Ablation studies confirm the effectiveness of each component in achieving stable, adaptive, and highly effective scheduling in dynamic cloud environments.

## 2 RELATED WORK

**Learning Approaches for Scheduling.** Learning approaches for scheduling generally follow two paradigms. **Improvement** methods iteratively refine existing full solutions via local search or priority adjustments (Pirnay & Grimm, 2024; Zhang et al., 2024a), which are ill-suited to dynamic environments. **Constructive** methods incrementally build up complete solutions by learning a direct mapping from state to scheduling actions (Corsini et al., 2024; Yang et al., 2025; Zhang et al., 2023), which are naturally suited for dynamic long-term systems.

Existing research has predominantly focused on offline scheduling in static environments (Corsini et al., 2024; Li et al., 2021; Smit et al., 2024; Song et al., 2022; Zhang et al., 2020), largely overlooking the necessity for continual adaptation in online settings. Online RL offers a viable framework for improving pre-trained schedulers without the need for costly retraining (Hamadanian et al., 2025; Sherman et al., 2023). However, prior scheduling works focused mainly on state representations (Song et al., 2022; Zhang et al., 2020) and network architectures (Shen et al., 2025; Wang et al., 2023), neglecting core algorithmic challenges, such as distribution shift, catastrophic forgetting, and learning stability. Although Yang et al. (2025) introduced an offline-online RL approach, its reliance on a single policy/actor may render it incompetent for robust, reliable online scheduling across a long time period. To address this gap, we propose a novel *robust actor-critic RL* approach with *ensemble techniques*, enhancing robustness, stability, and adaptive capacity for online DWS.

**Ensemble Methods in RL** Ensemble methods leverage multiple models to enhance overall performance and have been widely adopted in RL. They help mitigate approximation errors (An et al., 2021; Ball et al., 2023; Chen et al., 2021; Fujimoto et al., 2018), handle partial observability (Chen & Huang,

2023; Gao et al., 2024; Li et al., 2024), and balance the exploration–exploitation trade-off (Lee et al., 2021; Li et al., 2023; Lin et al., 2024). Based on their roles, these methods can be broadly categorized into ensembles of policies (i.e., actors) and ensembles of value functions (i.e., critics).

A central challenge in ensemble learning is how to effectively aggregate outputs from multiple networks. Common strategies for **action aggregation** include averaging action-selection probabilities $\pi(s, a)$ (Gao et al., 2024; Jiang et al., 2023), averaging actions $\pi(s)$ (Chen & Huang, 2023; Lin et al., 2024), or applying majority voting (Lin et al., 2024; Lyu et al., 2022; Osband et al., 2016). In discrete action spaces, voting is often preferred since relative selection frequency matters more than absolute selection probabilities. Lin et al. (2024) proposed to select actions receiving the highest votes, and breaks ties randomly. However, handling tied votes is particularly important because pre-trained policies frequently result in ties (See Figure 4 in Appendix A). The tie-breaking strategy is vital to ensuring high decision quality. Lee et al. (2021) used a Q-function to guide action selection and to encourage efficient exploration. Inspired by this idea, we introduce a *Value-ranked Action Aggregation* mechanism.

Critic ensembles are widely employed to reduce bias and variance in value estimation, thereby promoting robust, sample-efficient learning. For **value aggregation**, common approaches include averaging (Chen et al., 2021; Fang et al., 2025; Kostrikov et al., 2022), taking the minimum (Ball et al., 2023; Chen et al., 2025a; Fujimoto et al., 2018), or applying weighted or bounded combinations (Lee et al., 2021; Werge et al., 2025; Zhang et al., 2024b). However, these methods do not differentiate between critics with distinct adaptation speeds and roles. To address partial observability, Li et al. (2024) proposed a dual-critic setup consisting of an oracle critic with full state access and a standard critic limited to partial observations. While their goal was to leverage privileged information during training, our approach is motivated by a fundamentally different challenge of handling distribution shift in online RL. Drawing inspiration from the dual-critic concept, we propose *Dual Critic Ensembles with Decoupled Updates*, a new mechanism tailored for DWS. Specifically, we design two critic ensembles: one updates conservatively to preserve offline knowledge, and the other updates frequently to track changes in online observations.

**Continual and Online RL.** Recent advances in continual RL (CRL) tackle stability–plasticity trade-offs using regularization (Kirkpatrick et al., 2017), prompting (Wang et al., 2022), or reset-based strategies that preserve plasticity (Dohare et al., 2024; Farias & Jozefiak, 2025). In online RL, efficient fine-tuning (Zhou et al., 2025) and context-aware adaptation (Hamadanian et al., 2025) have shown promise. However, these approaches are largely designed for continuous control or multi-task settings, where small errors are recoverable. In contrast, combinatorial scheduling is highly unforgiving: a single suboptimal assignment can propagate through DAG dependencies and trigger system-wide delays. As a result, existing continual or online RL methods cannot be applied directly and require substantial redesign to function in this setting.

RAISE fills this gap with a novel ensemble architecture that maintains stability while adapting to non-stationary workload dynamics. To the best of our knowledge, it is among the first methods to bring principled Online RL to the NCO domain.

## 3 PRELIMINARY

**Dynamic Workflow Scheduling.** A problem instance of DWS is defined by a set $\mathcal{W}$ of workflows that arrive dynamically and a pool $\mathcal{M}$ of heterogeneous machines. Each workflow $W_i \in \mathcal{W}$ is represented as a *Directed Acyclic Graph* (DAG), denoted as $W_i = (\mathcal{O}_{W_i}, \mathcal{C}_{W_i})$, where $\mathcal{O}_{W_i} = \{O_{i1}, \ldots, O_{in_i}\}$ is the set of *tasks* and $\mathcal{C}_{W_i} \subseteq \mathcal{O}_{W_i} \times \mathcal{O}_{W_i}$ represents dependencies between tasks. An edge $(O_{ij}, O_{ik}) \in \mathcal{C}_{W_i}$ captures a task precedence constraint, indicating that task $O_{ij}$ must complete before task $O_{ik}$ can begin. The full set of all tasks across all workflows is denoted as $\mathcal{O} = \bigcup_i \mathcal{O}_{W_i}$.

At each decision step $t$, a single **focused task** $O_t^* \equiv O_{ij}$ is considered. Task $O_t^*$ is *ready* for execution since all its predecessors have completed. Because $O_t^*$ is unassigned, it is identified by the system and scheduled according to a policy. Each task $O_{ij}$ has a **workload** $tw_{ij} \in \mathbb{R}^+$ and can be assigned to any machine $M_q \in \mathcal{M}$. The **execution time** of task $O_{ij}$ on machine $M_q$ is given by $et_{ij}^{(q)} = tw_{ij}/ms_q$, where $ms_q$ is the **processing speed** of machine $M_q$. Let $st_{ij}^q$ and $ft_{ij}^q$ denote the **start time** and **finish time** of task $O_{ij}$ on machine $M_q$, we have $ft_{ij}^q = st_{ij}^q + et_{ij}^q$. When all the tasks $\mathcal{O}_{W_i}$ have finished, the **complete time** of workflow $W_i$ is calculated by $ft_i = \max_j(ft_{ij})$. Given its **arrival**

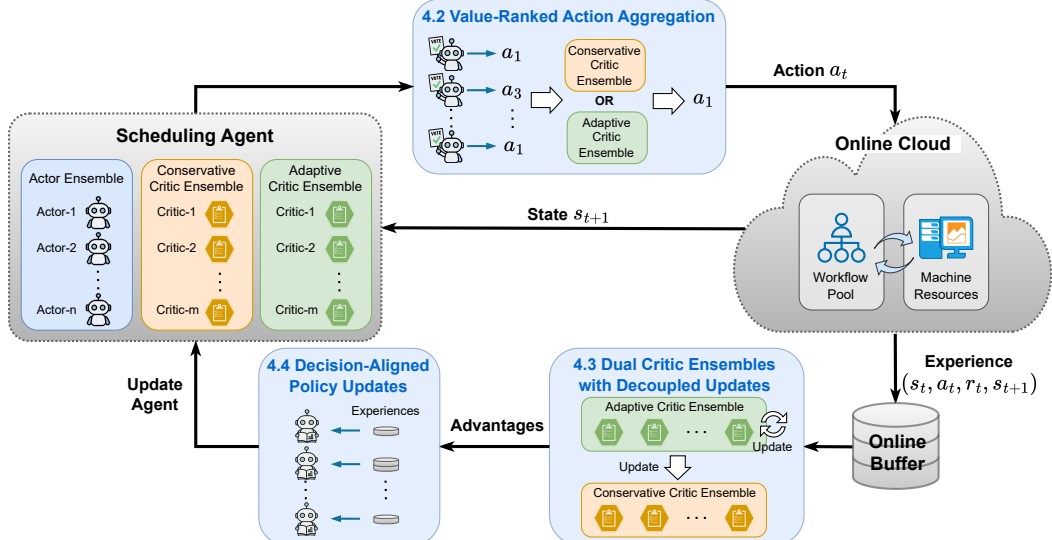

Figure 1: **The overall framework of RAISE for dynamic workflow scheduling.**

**time** as $at_i$, the **flowtime** of workflow $W_i$ is defined by $F_i = ft_i - at_i$. A detailed formulation of the scheduling process can be found in Figure 5, Appendix B.

Following a policy $\pi$, the scheduling agent assigns each focused task $O_t^* \in \mathcal{O}$ to a machine $M_q \in \mathcal{M}$. It is then placed in the machine's FIFO-based waiting queue. The ultimate **goal** is to learn a policy $\pi$ that minimizes the mean-flowtime: $\bar{F} = \frac{1}{|\mathcal{W}|} \sum_{i=1}^{|\mathcal{W}|} F_i$, while complying with task precedence constraints.

**Dynamic Graph Representation.** Recent learning-based approaches to DWS (Shen et al., 2025; Yang et al., 2025) leverage DAG-based dynamic graph representations to encode rich task, workflow, and machine information. We adopt a similar strategy (See Figure 6 and Figure 7 in Appendix C), where multiple workflows are merged into a global graph by adding edges that represent task execution order across machines. The graph evolves in real time: task nodes are removed once executed, and workflows are added upon arrival, with no visibility into future workloads. In line with actor-critic RL approaches, we adopt a system-oriented graph $\mathcal{G}^c$ for the critic to accurately estimate global value, and a task-specific graph $\mathcal{G}^a$ for the actor in action-conditioned policy scoring. See Appendix C for more details. We use Graph Attention Networks (GAT) (Veličković et al., 2018) as the feature extractor, due to their ability to handle heterogeneous relations and selectively attend to important node dependencies. Refer to Appendix D for detailed neural network architecture designs.

## 4 METHODOLOGY

We now present the core design of **RAISE** (**R**obust **A**ctor-Critic **I**ntegration for **S**cheduling **E**nsembles), our online RL approach with ensemble techniques for DWS. Particularly, RAISE leverages multiple pre-trained actors and critics that are continually improved through a PPO-based algorithm tailored for online learning from non-stationary input streams. Figure 1 illustrates three key components in our approach: (i) *value-ranked action aggregation* for stable and reliable decision-making, (ii) *dual critic ensembles with decoupled updates* to balance adaptation and value stability, and (iii) *decision-aligned policy updates* to improve learning efficiency while preserving ensemble diversity. The overall learning process alternates between executing scheduling actions and updating the actors and critics based on state-transition samples collected from the live environment. The detailed training procedure is presented in Algorithm 1 (see Appendix F).

A common concern with ensemble methods is computational overhead, especially in online settings that demand rapid decisions and continuous adaptation. RAISE is designed for efficiency: all actors and critics share a lightweight architecture and support fast parallel inference. Action selection involves only forward passes and simple ranking over a small candidate set, while updates are performed in a batched, actor-specific fashion. Empirically, RAISE achieves real-time performance on standard workflow traces with minimal overhead.

## 4.1 Problem Formulation and Ensemble Architecture

We adopt an RL formulation for DWS (Shen et al., 2025; Yang et al., 2025). First, the DWS environment is modeled as a *Markov Decision Process* (MDP) characterized by the tuple $(\mathcal{S}, \mathcal{A}, \mathcal{P}, \mathcal{R}, \gamma)$ with the following definitions:

- **State**. At each decision step $t$, the state $s_t \in \mathcal{S}$ captures all uncompleted tasks, machine statuses, and workflow information. The scheduling process is *event-driven*, and there is only one focused task $O_t^*$ waiting to be assigned at any decision step. Following Yang et al. (2025), we adopt two separate graph representations for the Actor-focused and Critic-focused states (see Appendix C), each tailored to the distinct roles of the Actor and Critic in AC-based RL. See Appendix E regarding all raw features utilized in our state representations.
- **Action**. An action $a_t \in \mathcal{A}$ indicates the assignment of the focused task $O_t^*$ to an available machine $M_q \in \mathcal{M}$.
- **Transition**. The system changes from state $s_t$ to state $s_{t+1}$ after taking action $a_t$, i.e., the assignment of task $O_t^*$ triggers the next decision step $t + 1$.
- **Reward**. If no workflows are completed during the two adjacent decision steps, $r_t = 0$; otherwise, $r_t = -\sum_{W_i \in \mathcal{W}^c} F_i$ gives the negative sum of the *flowtimes* of all workflows completed between decision steps $t$ and $t + 1$.

The goal of RAISE is to maximize the expected cumulative return within each operational cycle (e.g., a time window defined by a fixed number of completed workflows).

Specially, RAISE consists of $n$ **Actor** networks $\{\pi_{\theta_i}\}_{i=1}^n$, $m$ **Conservative Critic** networks $\{\hat{Q}_{\psi_k}\}_{k=1}^m$, and $m$ **Adaptive Critic** networks $\{Q_{\phi_j}\}_{j=1}^m$.

- **Conservative Critics** ($\hat{Q}_{\psi_k}$): The networks $\hat{Q}_{\psi_k}$ are infrequently updated to retain reliable value estimates from *prior knowledge* learned offline.
- **Adaptive Critics** ($Q_{\phi_j}$): The networks $Q_{\phi_j}$ are frequently updated using recent online experience, allowing them to adapt quickly to evolving system dynamics. $Q_{\phi_j}$ have identical trainable parameters as $\hat{Q}_{\psi_j}$ initially (i.e., $\forall j$, $\phi_j = \psi_j$) before online RL.

In our experiments, $m = n$. All actor and critic networks are pre-trained according to (Yang et al., 2025), before online improvement through RAISE.

## 4.2 Value-Ranked Action Aggregation

Traditional ensemble methods for action selection mainly rely on simple averaging (Jiang et al., 2023; Chen & Huang, 2023) or weighted voting (Wang et al., 2024). However, these techniques may perform poorly in large discrete action spaces or when actors have different levels of confidence. To address these issues, we propose a **two-stage** aggregation mechanism that combines policy consensus with value-based ranking for reliable action selection.

**Stage 1: Majority Voting.** Each actor $\pi_{\theta_i}$ independently selects its preferred action:

$$a_{t,i}^* = \arg\max_{a \in \mathcal{A}} \pi_{\theta_i}(a|s_t) \tag{1}$$

We then identify the action set $A_t^{tie} \subseteq \mathcal{A}$ that receives the most votes:

$$A_t^{tie} = \arg\max_{a \in \mathcal{A}} \sum_{i=1}^n \mathbb{I}[a_{t,i}^* = a] \tag{2}$$

If $|A_t^{tie}| = 1$, the only action in $A_t^{tie}$ is selected. Otherwise, we proceed to Stage 2.

**Stage 2: Value-Ranked Resolution.** When multiple actions receive the highest vote count in Stage 1, i.e., $|A_t^{tie}| > 1$, we resolve the tie using critic-derived ranks instead of raw Q-values

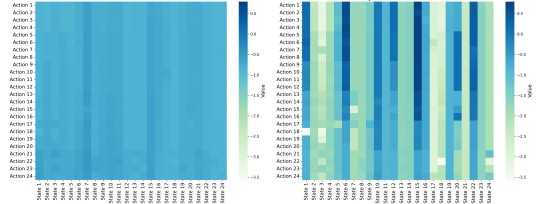

Figure 2: **Q-value distributions of Conservative (Left) and Adaptive (Right) Critic Ensembles on the same set of states and actions.**

(which may have different scales across critics). We randomly choose the set of Adaptive Critics $\{Q_{\phi_j}\}_{j=1}^m$ or the set of Conservative Critic $\{\hat{Q}_{\psi_k}\}_{k=1}^m$ to perform the ranking. This random switching prevents the final action selection from consistently favoring one critic set over the other. The adaptive critics may offer more accurate estimates but are less stable. The conservative critics provide stability

but tend to yield biased action value estimates due to distribution shifts. To validate these distinct behaviors, we visualize the value distributions of both ensembles during online adaptation in Figure 2. More analysis can be seen in Appendix J and Appendix M.

For each action $a \in A_t^{tie}$ and each critic $Q_{\phi_j}$ or $\hat{Q}_{\psi_k}$, we compute its rank by:

$$\text{rank}_j(a|s_t) = \#\{a' \in A_t^{tie} : Q_{\phi_j}(s, a') > Q_{\phi_j}(s, a)\} \tag{3}$$

$$\text{rank}_k(a|s_t) = \#\{a' \in A_t^{tie} : \hat{Q}_{\psi_k}(s, a') > \hat{Q}_{\psi_k}(s, a)\} \tag{4}$$

The final action is selected based on the average rank:

$$a_t = \arg\min_{a \in A_t^{tie}} \frac{1}{m} \sum_{j=1}^m \text{rank}_j(a|s_t) \quad \text{or} \quad a_t = \arg\min_{a \in A_t^{tie}} \frac{1}{m} \sum_{k=1}^m \text{rank}_k(a|s_t) \tag{5}$$

This approach relies on the relative ranking of actions instead of their absolute value estimates, making the selection process more robust to scale differences and estimation biases across critics. As shown in the ablation study in Section 5.3, our value-ranked method outperforms both value-based (Sun et al., 2021) and probability-based (Jiang et al., 2023) action aggregation strategies. Empirical analysis confirms that VRAA does not cause extremely imbalance in actor selection (see Appendix K).

## 4.3 DUAL CRITIC ENSEMBLES WITH DECOUPLED UPDATES

To balance exploration and exploitation, we train Adaptive Critics $Q_{\phi_j}$ to quickly track recent experiences. Meanwhile, Conservative Critics $\hat{Q}_{\psi_k}$ preserve stable value estimates from prior knowledge.

Each Adaptive Critic $Q_{\phi_j}$ is updated by minimizing the Mean Squared Error (MSE) loss against the Monte Carlo return with a high Update-to-Data (UTD) ratio (Werge et al., 2025):

$$\mathcal{L}^Q(\phi_j) = \frac{1}{2} \mathbb{E}_t[(Q_{\phi_j}(s_t, a_t) - \sum_{l=0}^{T-t} \gamma^l r_{t+l})^2] \tag{6}$$

The Conservative Critics are updated subsequently based on the Adaptive Critics via Polyak averaging:

$$\psi_k \leftarrow \tau\phi_j + (1 - \tau)\psi_k, \quad \forall j = k \tag{7}$$

where $0 < \tau \ll 1$ controls the update speed, producing stable and conservative targets for both action selection and policy learning, while gradually incorporating new information. Ablation study in Section 5.4 demonstrates the effectiveness of using dual critic ensembles.

## 4.4 DECISION-ALIGNED POLICY UPDATES

To support stable learning, we introduce a *selective update* mechanism described below. An ablation study is conducted in Section 5.4 to verify its effectiveness.

**Experience Assignment.** Each collected state-transition sample $(s_t, a_t, r_t, s_{t+1})$ is assigned only to the relevant actors that recommend the executed action $a_t$ at state $s_t$. If multiple actors selected $a_t$, the transition is added to each of their replay buffers. This ensures that actors are always trained on relevant data that reflects their individual action preferences.

**Policy Optimization.** Each actor $\pi_{\theta_i}$ is updated using the PPO-Clip objective, with advantages computed from the Adaptive Critic ensemble $\{Q_{\phi_j}\}_{j=1}^m$, which offers timely value estimates to guide online RL:

$$\mathcal{L}^\pi(\theta_i) = \mathbb{E}_{s_t \sim \mathcal{D}_i} \left[ \min \left( \frac{\pi_{\theta_i}(a_t|s_t)}{\pi_{\theta_{i,old}}(a_t|s_t)} \bar{A}_t, \text{clip} \left( \frac{\pi_{\theta_i}(a_t|s_t)}{\pi_{\theta_{i,old}}(a_t|s_t)}, 1 - \epsilon, 1 + \epsilon \right) \bar{A}_t \right) \right], \tag{8}$$

where $\bar{A}_t = \frac{1}{n} \sum_{j=1}^n (Q_{\phi_j}(s_t, a_t) - \sum_{l=0}^{T-t} \gamma^l r_{t+l})$.

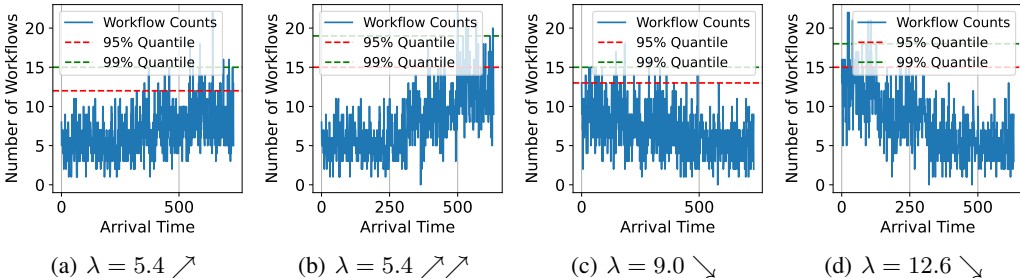

(a) $\lambda = 5.4 \nearrow$     (b) $\lambda = 5.4 \nearrow\nearrow$     (c) $\lambda = 9.0 \searrow$     (d) $\lambda = 12.6 \searrow\searrow$

Figure 3: **Arrival counts within the 95%–99% quantile range across scenarios with different arrival rate changes.**

**Gradient Control.** To prevent catastrophic forgetting and ensure stable online adaptation, we apply a conservative form of gradient clipping while training actor networks:

$$\nabla_{\theta_i} \mathcal{L}^\pi = \begin{cases} \nabla_{\theta_i} \mathcal{L}^\pi, & \text{if } \|\nabla_{\theta_i} \mathcal{L}^\pi\|_2 \leq \min(\rho_{\text{prev}} + \sigma_{\text{prev}}, \tau_0), \\ \mathbf{0}, & \text{otherwise,} \end{cases} \tag{9}$$

where $\mu_{prev}$ and $\sigma_{prev}$ are the mean and standard deviation of gradient norms from the previous batch. The effective threshold is defined as $\min(\rho_{\text{prev}} + \sigma_{\text{prev}}, \tau_0)$, where $\tau_0$ is the maximum allowed gradient scale, serving as a hard safety cap to prevent gradient explosion. Instead of rescaling large gradients, we discard them when they exceed $\tau_0$. This conservative strategy avoids propagating unstable updates that may quickly degrade online performance in non-stationary environments. Based on our experiments, we recommend setting $\tau_0$ to the mean plus standard deviation of gradient norms observed during the offline pre-training phase.

## 5 EXPERIMENTS

### 5.1 ENVIRONMENTAL SETTINGS

**Workflow Patterns.** Following common practice in prior workflow scheduling research (Qin et al., 2023; Xu et al., 2023; Shen et al., 2024; Yang et al., 2025), we simulate scheduling workloads using four real-world scientific workflow patterns: Montage, CyberShake, SIPHT, and Inspiral. Each workflow pattern is provided as a DAG with metadata (e.g., number of tasks, number of edges, per-task workload) available from the Pegasus[1] repository (Deelman et al., 2015) (see Appendix G.1 for detailed information). The four workflow classes exhibit high heterogeneity in total workload (up to $\approx 30 \times$ difference), producing realistic diversity in scheduling difficulty (Sun et al., 2024).

**Machine Configurations.** We consider a heterogeneous cloud environment involving six machine types based on Amazon EC2[2] instances. Each type varies in multiple dimensions, such as processing speed and price (see Table 6 in Appendix G.2). We adopt mixed pools of machines, indicated by "*number of type × number of amount*". For example, "$4 \times 10$" means the first 4 machine types in Table 6, with 10 instances of each. In this paper, we evaluate **three** mixed pools: "$6 \times 4$", "$5 \times 7$", and "$4 \times 10$".

**Arrival Patterns.** Consistent with prior studies (Wang et al., 2019; Shen et al., 2024; Sun et al., 2024), workflow arrivals are modeled as a Poisson process. Unlike previous work that uses a single fixed arrival rate ($\lambda$), we use **time-varying** arrival rates to test online adaptation under *non-stationary* conditions. We define **four** patterns with arrows representing increasing/decreasing $\lambda$: "$5.4 \nearrow$" (increases from 5.4 to 9.0 workflows/hour), "$5.4 \nearrow\nearrow$" (increases from 5.4 to 12.6 workflows/hour), "$9.0 \searrow$" (decreases from 9.0 to 5.4 workflows/hour), and "$12.6 \searrow$" (decreases from 12.6 to 5.4 workflows/hour). See Figure 3 for the quantile plots of the variations in arrival rates.

**Online Scenarios.** Combining the **three** mix machine pools with the **four** arrival patterns yields $3 \times 4 = 12$ online scenarios. Each scenario involves scheduling 20,000 workflows, amounting to approximately **600,000 real-time decisions**. A scenario is labeled as $\langle$*MachineConfig, ArrivalRate, WorkflowCount*$\rangle$, e.g., $\langle 4 \times 10, 5.4 \nearrow\nearrow, 20k \rangle$ means 20k workflows with arrival rates $\lambda$ increasing from 5.4 to 12.6 workflows/hour need to be allocated to the machine pool of $4 \times 10 = 40$ machines.

---

[1]https://download.pegasus.isi.edu/misc/SyntheticWorkflows.tar.gz
[2]https://aws.amazon.com/ec2/pricing/on-demand/

**Baselines.** We compare our proposed method, **RAISE**, against a range of strong baselines (see detailed implementation in Appendix H):

- *Expert-designed PDRs*: **EST** (Chetto & Chetto, 1989), **PEFT** (Senapati et al., 2021), and **HEFT** (Topcuoglu et al., 2002) are three widely-used manual priority dispatching rules (PDRs).

- *Tree-based PDRs*: **GPHH** (Xu et al., 2023) is a state-of-the-art hyper-heuristic for automatically evolving tree-based PDRs for DWS. We report the best-performing heuristic obtained by GPHH from 30 independent runs.

- *Neural PDRs*: **ERL-DWS** (Shen et al., 2024) is a policy-based RL approach with a Transformer-based architecture; and **GOODRL** (Yang et al., 2025) is an offline-online RL approach with a single GAT-based agent. We report their best performance among 5 random seeds.

All learning-based baselines (i.e., GPHH, ERL-DSW, GOODRL) are trained offline on the scenario $\langle 5 \times 5, \ 5.4, \ 30 \rangle$ to exercise their ability to adapt to *non-stationary* workload in the online settings. For fair comparison, RAISE is initialized with an ensemble of 5 actors, 5 conservative critics, and 5 adaptive critics, using pre-trained GOODRL models from the same offline scenario $\langle 5 \times 5, \ 5.4, \ 30 \rangle$. As online learning approaches, Neural PDRs controlled by GOODRL and RAISE are continuously fine-tuned in online scenarios.

**Hyperparameters.** Our actor network employs a 2-layer GAT followed by a 4-layer MLP. The critic network employs a 2-layer GAT, a 1-layer self-attention mechanism, and a 4-layer MLP, all with a hidden dimension of 128. Further details regarding network architecture, RAISE configuration, and computing environment are provided in Appendix I. Scalability analysis in Appendix Q justifies our choice of $n = 5$ as the optimal trade-off ensemble size.

**Metrics.** To quantify online adaptation, we report (1) *mean-flowtime* (obj.) defined by $\bar{F} = \frac{1}{|\mathcal{W}|} \sum_{i=1}^{|\mathcal{W}|} F_i$; and (2) *Relative Improvement* defined as follows:

$$\Delta_\% = 100 \times \frac{\rho_{\text{base}} - \rho_{\text{algo}}}{\rho_{\text{base}} + \epsilon} \tag{10}$$

where $\rho_{\text{base}} = \frac{1}{|\mathcal{W}|} \sum_{i=1}^{|\mathcal{W}|} \frac{F_i^{\text{base}}}{l_i}$, $\rho_{\text{algo}} = \frac{1}{|\mathcal{W}|} \sum_{i=1}^{|\mathcal{W}|} \frac{F_i^{\text{algo}}}{l_i}$, and workload of $W_i$ is $l_i = \sum_j tw_{ij}$. The equation 10 represents the percentage reduction in flowtime per workload unit compared to a baseline (e.g., HEFT) at identical interaction steps. A positive value indicates superior adaptation efficiency.

## 5.2 Online Performance

Table 1 presents the *mean-flowtime* results of all algorithms across 12 online scenarios, covering all combinations of three mixed machine pools and four arrival patterns. Overall, our ensemble-based method **RAISE** achieves the best performance on most of the scenarios.

Among baselines, *expert-designed PDRs* (i.e., EST, PEFT, HEFT) show significant performance gaps from 18.36% to 254.21%, due to their inability to adapt to non-stationary conditions. GPHH demonstrates competitive performance in some scenarios but exhibits instability with gaps up to 5.35%. It should be noted that GPHH's result is the best of 30 runs, whereas RAISE's performance is averaged over 5 random seeds. ERL-DWS performs substantially worse than RAISE in all cases. As a strong DRL baseline, GOODRL delivers competitive performance, but RAISE consistently surpasses it by margins of up to 6.10%, underscoring the benefits of our ensemble-based online learning approach.

Although RAISE incurs slightly longer inference time (i.e., 0.0341-0.0864s) compared to rule-based methods (i.e., 0.0001-0.0010s), its decision latency remains significantly lower than typical data transfer and communication delays in real-world cloud environments (often ranging from seconds to minutes). This makes the computational overhead practically negligible and ensures the feasibility of deployment in dynamic environments with time-varying workloads. Furthermore, our energy efficiency analysis demonstrates that the computational overhead of RAISE is negligible ($< 0.1\%$) compared to the energy savings achieved by reduced workflow flowtime (see Appendix L).

These results demonstrate the robustness and adaptability of RAISE across varying machine configurations and workflow arrival patterns, highlighting its suitability for practical DWS applications.

Table 1: **Performance comparison in online scenarios.** "Obj.": the *mean-flowtime* across 20,000 workflows. "Gap": the gap to the best "Obj." in each scenario. "Time(s)": the average inference time of making a decision in CPU. "**bold**": the best result in each scenario.

| Methods | $\langle 6 \times 4, 5.4 \nearrow, 20k \rangle$ | | | $\langle 6 \times 4, 5.4 \nearrow\nearrow, 20k \rangle$ | | | $\langle 6 \times 4, 9.0 \searrow, 20k \rangle$ | | | $\langle 6 \times 4, 12.6 \searrow, 20k \rangle$ | | |
|---|---|---|---|---|---|---|---|---|---|---|---|---|
| | Obj.↓ | Gap↓ | Time(s)↓ | Obj.↓ | Gap↓ | Time(s)↓ | Obj.↓ | Gap↓ | Time(s)↓ | Obj.↓ | Gap↓ | Time(s)↓ |
| EST | 1037.70 | 254.21% | 0.0001 | 999.81 | 235.16% | 0.0001 | 1029.77 | 259.04% | 0.0002 | 998.66 | 241.82% | 0.0003 |
| PEFT | 413.43 | 41.12% | 0.0006 | 400.25 | 34.18% | 0.0008 | 412.10 | 43.68% | 0.0004 | 398.61 | 36.43% | 0.0005 |
| HEFT | 371.57 | 26.83% | 0.0007 | 363.34 | 21.80% | 0.0004 | 371.74 | 29.61% | 0.0002 | 363.83 | 24.53% | 0.0004 |
| GPHH | 296.57 | 1.23% | 0.0052 | **298.30** | **0.00%** | 0.0071 | 296.67 | 3.44% | 0.0053 | 298.29 | 2.10% | 0.0038 |
| ERL-DWS | 2867.11 | 878.65% | 0.0034 | 3967.00 | 1229.85% | 0.0034 | 2190.09 | 663.60% | 0.0039 | 2166.48 | 641.54% | 0.0035 |
| GOODRL | 294.46 | 0.51% | 0.0177 | 300.81 | 0.84% | 0.0070 | 299.46 | 4.41% | 0.0167 | 309.99 | 6.10% | 0.0063 |
| **RAISE** | **292.96** | **0.00%** | 0.0444 | 311.31 | 4.36% | 0.0627 | **286.81** | **0.00%** | 0.0341 | **292.16** | **0.00%** | 0.0390 |

| Methods | $\langle 5 \times 7, 5.4 \nearrow, 20k \rangle$ | | | $\langle 5 \times 7, 5.4 \nearrow\nearrow, 20k \rangle$ | | | $\langle 5 \times 7, 9.0 \searrow, 20k \rangle$ | | | $\langle 5 \times 7, 12.6 \searrow, 20k \rangle$ | | |
|---|---|---|---|---|---|---|---|---|---|---|---|---|
| | Obj.↓ | Gap↓ | Time(s)↓ | Obj.↓ | Gap↓ | Time(s)↓ | Obj.↓ | Gap↓ | Time(s)↓ | Obj.↓ | Gap↓ | Time(s)↓ |
| EST | 1303.53 | 234.18% | 0.0003 | 1276.15 | 223.66% | 0.0002 | 1296.81 | 235.61% | 0.0002 | 1275.75 | 227.75% | 0.0004 |
| PEFT | 537.21 | 37.72% | 0.0005 | 522.26 | 32.46% | 0.0006 | 536.67 | 38.89% | 0.0003 | 521.92 | 34.08% | 0.0007 |
| HEFT | 495.04 | 26.91% | 0.0001 | 484.25 | 22.82% | 0.0005 | 494.50 | 27.98% | 0.0003 | 484.83 | 24.56% | 0.0005 |
| GPHH | 397.48 | 1.90% | 0.0054 | 399.57 | 1.34% | 0.0076 | 397.35 | 2.83% | 0.0038 | 399.54 | 2.65% | 0.0041 |
| ERL-DWS | 2479.09 | 535.55% | 0.0032 | 2918.95 | 640.31% | 0.0049 | 2001.72 | 418.04% | 0.0031 | 1955.07 | 402.27% | 0.0040 |
| GOODRL | 396.41 | 1.62% | 0.0200 | 403.20 | 2.26% | 0.0058 | 400.30 | 3.60% | 0.0120 | 400.06 | 2.78% | 0.0144 |
| **RAISE** | **390.07** | **0.00%** | 0.0603 | **394.28** | **0.00%** | 0.0640 | **386.40** | **0.00%** | 0.0764 | **389.24** | **0.00%** | 0.0758 |

| Methods | $\langle 4 \times 10, 5.4 \nearrow, 20k \rangle$ | | | $\langle 4 \times 10, 5.4 \nearrow\nearrow, 20k \rangle$ | | | $\langle 4 \times 10, 9.0 \searrow, 20k \rangle$ | | | $\langle 4 \times 10, 12.6 \searrow, 20k \rangle$ | | |
|---|---|---|---|---|---|---|---|---|---|---|---|---|
| | Obj.↓ | Gap↓ | Time(s)↓ | Obj.↓ | Gap↓ | Time(s)↓ | Obj.↓ | Gap↓ | Time(s)↓ | Obj.↓ | Gap↓ | Time(s)↓ |
| EST | 1513.03 | 192.79% | 0.0002 | 1493.18 | 188.49% | 0.0002 | 1509.89 | 199.64% | 0.0004 | 1489.85 | 196.18% | 0.0005 |
| PEFT | 663.22 | 28.34% | 0.0010 | 648.62 | 25.32% | 0.0008 | 662.29 | 31.43% | 0.0007 | 648.45 | 28.91% | 0.0004 |
| HEFT | 623.91 | 20.74% | 0.0004 | 612.63 | 18.36% | 0.0003 | 623.46 | 23.72% | 0.0002 | 613.44 | 21.95% | 0.0001 |
| GPHH | 516.89 | 0.03% | 0.0040 | 528.03 | 2.02% | 0.0039 | 517.38 | 2.67% | 0.0057 | 529.94 | 5.35% | 0.0073 |
| ERL-DWS | 2696.87 | 421.88% | 0.0049 | 3090.58 | 497.12% | 0.0029 | 2235.61 | 343.65% | 0.0050 | 2154.00 | 328.21% | 0.0049 |
| GOODRL | 523.88 | 1.38% | 0.0200 | 532.80 | 2.94% | 0.0096 | 522.23 | 3.64% | 0.0201 | 530.18 | 5.40% | 0.0188 |
| **RAISE** | **516.76** | **0.00%** | 0.0587 | **517.58** | **0.00%** | 0.0864 | **503.91** | **0.00%** | 0.0515 | **503.02** | **0.00%** | 0.0843 |

## 5.3 ABLATION STUDY OF ACTION AGGREGATION METHODS

To evaluate the effectiveness of the proposed *Value-Ranked Action Aggregation* (VRAA) mechanism, we conduct an ablation study to compare four action selection strategies across six scenarios. To accurately analyze the effect of action selection, all mechanisms are tested directly without online adaptation. We compare (i) **Single actor**: Only a single actor/policy network is used for action selection, the same as in GOODRL (Yang et al., 2025); (ii) **Ensemble with prob.**: The action is selected by averaging the probabilities across all actor networks, $\frac{1}{n} \sum_{i=1}^{n} \pi_i(s, a)$ (Jiang et al., 2023); (iii) **Ensemble with value**: The action is chosen by applying majority voting, followed by using the average Q-values across all critics, $\frac{1}{m} \sum_{j=1}^{m} Q_j(s, a)$ (Lee et al., 2021); and (iv) **Ensemble with VRAA (Ours)**: The action is selected based on value-ranked aggregation, where average critic rankings are used to resolve ties after majority voting. For the **Single Actor** baseline, we report the results from the best-performing actor among five random seeds. For other approaches, we use all five seeds to build the ensemble. To isolate the effectiveness of the aggregation logic, both methods utilize the same set of pre-trained Conservative Critics.

Table 2 reports the *mean-flowtime* across 20,000 workflows and the Gap to the best method in each scenario. Table 2 demonstrates that **Ensemble with VRAA** achieves the best performance in 4 out of 6 scenarios, particularly in large-scale cluster environments (i.e., the machine pool of $4 \times 10 = 40$ machines). **Ensemble with value** shows competitive results in one scenario but exhibits instability in others. This is because absolute Q-values vary in scale across critics and are therefore unreliable, whereas the relative rankings used in VRAA provide more stable and informative measures of action quality. This rationale also explains why **Ensemble with prob.** performs worse than VRAA. Although the **Single Actor** approach is computationally efficient, it struggles to adapt to workload changes under dynamic conditions.

## 5.4 ABLATION STUDY OF ACTOR-CRITIC UPDATING METHODS

To assess the contribution of key components in RAISE, we conduct an ablation study to compare the full RAISE method against its two variants: (i) **RAISE w/o. Dual-Q**, which removes the dual critic

Table 2: **Ablation results on action aggregation methods.** "Obj.": the *mean-flowtime* across 20,000 workflows without fine-tuning. "Gap": the gap to the best "Obj." in each scenario.

| Mechanisms | $\langle 6 \times 4, 5.4, 20k \rangle$ Obj. | Gap | $\langle 5 \times 7, 5.4, 20k \rangle$ Obj. | Gap | $\langle 4 \times 10, 5.4, 20k \rangle$ Obj. | Gap |
|---|---|---|---|---|---|---|
| Single actor (Yang et al., 2025) | **285.55** | **0.00** | 389.75 | 2.17 | 513.40 | 2.60 |
| Ensemble with prob. (Jiang et al., 2023) | 289.19 | 3.64 | 388.45 | 0.87 | 523.80 | 13.00 |
| Ensemble with value (Lee et al., 2021) | 288.24 | 2.69 | 387.89 | 0.31 | 512.48 | 1.68 |
| **Ensemble with VRAA (Ours)** | 287.78 | 2.23 | **387.58** | **0.00** | **510.80** | **0.00** |
| Mechanisms | $\langle 6 \times 4, 9.0, 20k \rangle$ Obj. | Gap | $\langle 5 \times 7, 9.0, 20k \rangle$ Obj. | Gap | $\langle 4 \times 10, 9.0, 20k \rangle$ Obj. | Gap |
| Single actor (Yang et al., 2025) | 301.44 | 1.01 | 396.92 | 2.74 | 529.69 | 13.30 |
| Ensemble with prob. (Jiang et al., 2023) | 303.67 | 3.24 | 394.28 | 0.09 | 524.04 | 7.65 |
| Ensemble with value (Lee et al., 2021) | **300.43** | **0.00** | 394.45 | 0.27 | 522.51 | 6.12 |
| **Ensemble with VRAA (Ours)** | 300.89 | 0.47 | **394.18** | **0.00** | **516.38** | **0.00** |

Table 3: **Ablation results on actor-critic updating methods.** "**bold**": the best result in each scenario.

| Baselines | $\langle 6 \times 4, 5.4, 20k \rangle$ | $\langle 6 \times 4, 9.0, 20k \rangle$ | $\langle 5 \times 7, 5.4, 20k \rangle$ | $\langle 5 \times 7, 9.0, 20k \rangle$ | $\langle 4 \times 10, 5.4, 20k \rangle$ | $\langle 4 \times 10, 9.0, 20k \rangle$ |
|---|---|---|---|---|---|---|
| RAISE w/o. Dual-Q | 287.85 | 300.39 | 406.05 | 410.22 | 542.69 | 558.12 |
| RAISE w/o. Grad. | 289.27 | 352.13 | 471.30 | 557.45 | 515.39 | 764.79 |
| **RAISE** | **285.70** | **299.76** | **388.71** | **390.11** | **509.09** | **516.66** |

ensembles with decoupled updates (in Section 4.3) and uses only the adaptive critic for both action aggregation and policy updates; and (ii) **RAISE w/o. Grad.**, which removes the decision-aligned sample assignment and gradient control mechanism (in Section 4.4).

Table 3 reports the online performance, measured by *mean-flowtime*, across 20,000 workflows in six scenarios. Our **RAISE** method in this table achieves the best performance across all six scenarios. **RAISE w/o. Dual-Q** shows consistently worse performance than RAISE, indicating that it is critical to adopt the decoupled dual-critic design so as to effectively balance the trade-off between new experiences and prior knowledge. **RAISE w/o. Grad.** also exhibits significant performance drops compared to RAISE, e.g., objective increases from 390.1 to 557.45 in scenario $\langle 5 \times 7, 9.0, 20k \rangle$. This drastic drop indicates that the training process becomes unstable without our gradient-controlled policy updates. Regarding sample efficiency, comparisons in Appendix N demonstrate that RAISE achieves $\sim 2\times$ superior adaptation at identical interaction steps compared to single-policy baselines.

Table 4: **Ablation results on advantage calculation.** Relative improvement over HEFT.

| Advantage | 60-th | 80-th | 100-th | 120-th | 140-th | 160-th |
|---|---|---|---|---|---|---|
| **Adaptive (Ours)** | 7.99% | 12.58% | 16.99% | 20.18% | 22.26% | 23.42% |
| Conservative | 7.76% | 13.49% | 13.74% | 10.93% | 7.69% | 5.36% |
| Both | 6.84% | 11.36% | 12.87% | -17.80% | – | – |

Table 4 validates using Adaptive Critics for advantage calculation (equation 8). Using Adaptive Critics achieves the highest relative improvement (23.42%), whereas using Conservative Critics results in slower adaptation (degrading to 5.36%). The Both strategy (averaging) performs poorly, leading to divergence. This confirms the effectiveness of using Adaptive Critics in policy updates.

## 6 Conclusion and Future Work

This paper introduced *Robust Actor-Critic Integration for Scheduling Ensembles* (**RAISE**), an ensemble-based online RL approach for DWS. Unlike prior methods that rely on a single policy network, RAISE leverages multiple actors and critics, and incorporates three key technical innovations: (i) a value-ranked action aggregation mechanism that leverages relative critic rankings for robust decision-making; (ii) a dual-critic updating mechanism that balances fast adaptation to online data with the stable retention of offline knowledge; and (iii) a decision-aligned policy update technique to preserve ensemble diversity and prevent destructive policy updates. Experimental results demonstrated that RAISE can significantly outperform state-of-the-art baselines across diverse online scenarios, achieving superior adaptability to varying workload patterns and machine configurations.

Several directions remain for future study. RAISE can be extended to handle hard deadlines or budget constraints, and to support multi-objective optimization. It is also valuable to evaluate RAISE on production systems in order to fully assess its economic value and potential for real-world deployment.

## THE USE OF LARGE LANGUAGE MODELS

During the preparation of this manuscript, we made limited use of large language models (LLMs) to refine the writing (e.g., grammar, clarity, and readability) and to generate small logos (i.e., robots), which were subsequently used to draw figures by the authors. Importantly, all conceptual contributions, technical methods, analyses, and experimental results are original and developed entirely by the authors. The authors have thoroughly verified the correctness of all claims and remain fully responsible for the content of this paper, in accordance with the ICLR Code of Ethics.

## ETHICS STATEMENT

This work focuses on reinforcement learning methods for dynamic workflow scheduling in cloud computing environments. It does not involve human subjects, personal data, or sensitive information. The datasets used are synthetically generated based on publicly available benchmarks and do not contain personally identifiable or proprietary data. The research does not raise direct concerns related to fairness, privacy, or security. All authors have adhered to the ICLR Code of Ethics throughout the research and submission process.

## REPRODUCIBILITY STATEMENT

We have made every effort to ensure the reproducibility of our results. The problem formulation, algorithm details, and training procedures are clearly described in Sections 4 and Section 5.1. Hyperparameters and experimental settings are reported in Appendix I. Pseudocode of the proposed algorithm is provided in Algorithm 1. The full code for reproducing the reported results will also be made publicly available.

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

## A  TIE-BREAKERS IN MAJORITY VOTING

Figure 4 reports the percentage of tied votes in different online scenarios. We observe that ties occur frequently, with proportions ranging from about 16.9% to 23.1% across scenarios. Interestingly, scenarios with fewer machine pool sizes (e.g., $6 \times 4$) generally exhibit lower tie rates, while larger and more heterogeneous machine settings (e.g., $4 \times 10$) tend to produce higher tie rates. This indicates that as the action space expands and workload complexity increases, different actors in the ensemble are more likely to disagree, leading to tied votes. Such prevalence highlights the importance of designing principled tie-breaking strategies, as naive random selection could degrade decision quality and stability.

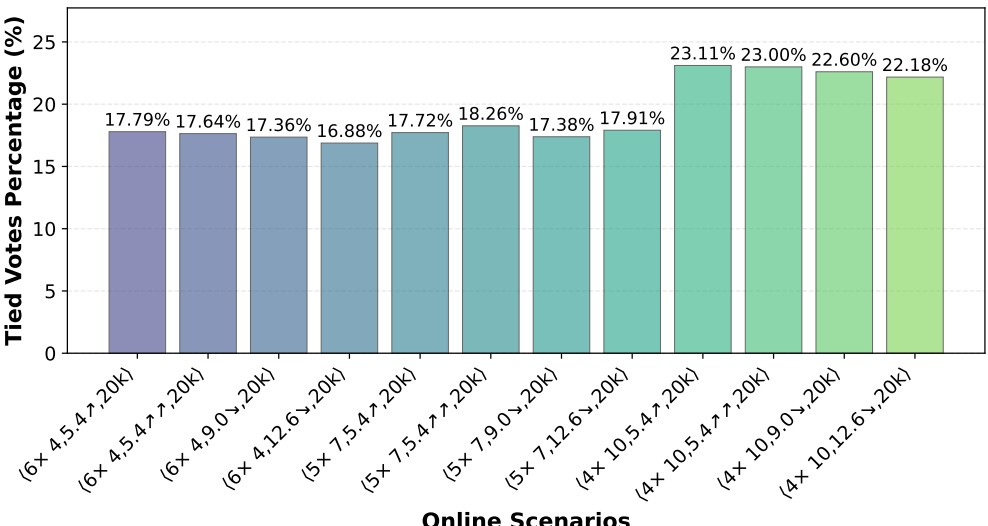

Figure 4: **Tied votes performance comparison across different online scenarios.**

## B  DWS PROBLEM FORMULATION

Figure 5 illustrates the scheduling process of dynamic workflow scheduling in a cloud system. Workflows dynamically arrive and are maintained in the workflow pool. Whenever the predecessor tasks of a node are completed, that task becomes a ready task as defined by equation 11. At each decision point, only one *focused task* is automatically identified by the cloud system as eligible for scheduling. If a focused task exists, the scheduler allocates it to one of the available machines; otherwise, the system waits for further task completions or workflow arrivals. Once a task is dispatched, it is placed in the queue of the selected machine and executed according to the First-In-First-Out (FIFO) rule. The workflow pool is continuously updated with the latest task completions, enabling subsequent tasks to become schedulable. This iterative process continues until all workflow tasks are completed.

The scheduling process adheres to the following constraints (Shen et al., 2025; Jayanetti et al., 2024; Xu et al., 2023):

- Only tasks whose all predecessors have been completed are eligible for scheduling.
- Each task must be assigned to exactly one VM and executed without interruption.
- Each VM can process at most one task at a time.
- Each task can be allocated to any available virtual machine.
- Task execution times vary across VMs due to heterogeneous computational capacities.

A task becomes *ready* when all its predecessors are completed. The ready time $rt_{ij}$ of task $O_{ij}$ is defined as:

$$rt_{ij} = \begin{cases} at_i, & \text{if } O_{ij} \text{ is an entry task} \\ \max_{O_{ik} \in \text{pred}(O_{ij})} \{ft_{ik}\}, & \text{otherwise} \end{cases} \quad (11)$$

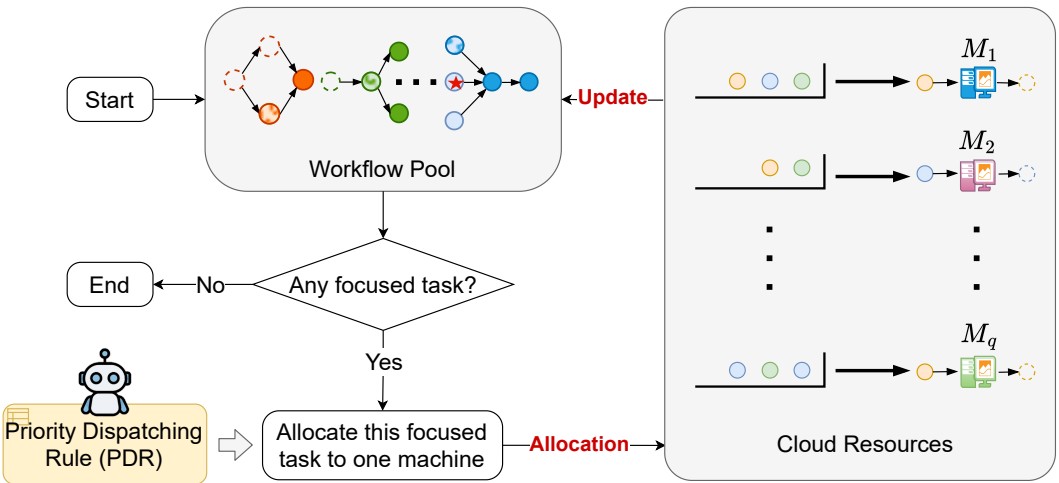

Figure 5: **Scheduling process of dynamic workflow scheduling in cloud computing.**

where $at_i$ is the arrival time of workflow $W_i$, $\text{pred}(O_{ij})$ denotes the set of predecessor tasks of $O_{ij}$, and $ft_{ik}$ is the finish time of task $O_{ik}$.

The execution time of task $O_{ij}$ on VM $M_q$ is given by:

$$et_{ij}^{(q)} = \frac{tw_{ij}}{ms_q} \tag{12}$$

where $tw_{ij}$ is the task workload and $ms_q$ is the computational capacity of $M_q$.

The finish time of task $O_{ij}$ on $M_q$ is:

$$ft_{ij}^{(q)} = st_{ij}^{(q)} + et_{ij}^{(q)} \tag{13}$$

where $st_{ij}^{(q)}$ is its start time on $M_q$. The waiting time of task $O_{ij}$ is defined as the time interval between its actual start time on machine $M_q$ and its ready time $rt_{ij}$:

$$wt_{ij}^{(q)} = st_{ij}^{(q)} - rt_{ij} \tag{14}$$

The workflow finish time of $W_i$ is:

$$ft_i = \max_{O_{ij} \in \mathcal{O}_{W_i}} \{ft_{ij}\} \tag{15}$$

The flowtime of workflow $W_i$ is $F_i = ft_i - at_i$. The objective is to find a scheduling policy $\pi$ that minimizes the mean-flowtime:

$$\bar{F} = \frac{1}{|\mathcal{W}|} \sum_{i=1}^{|\mathcal{W}|} F_i \tag{16}$$

subject to the precedence and resource constraints stated above.

**Online decision model.** In the online setting decisions are made sequentially at discrete decision steps $t$. At each step, a focused task $O_t^*$ (a ready, unassigned task) must be immediately assigned to a machine $M_q \in \mathcal{M}$. An assignment policy $\pi$ maps the current system state $s_t$ (including ready tasks, machine queues, remaining workloads, and recent arrivals) to an action $a_t \in \mathcal{A}$, i.e. $a_t \sim \pi(\cdot \mid s_t)$. The system then evolves as tasks execute and new workflows arrive. In practice, policies are commonly pretrained on historical traces and then fine-tuned online. Such online constraint motivates algorithmic considerations for stability, sample efficiency, and retention vs. adaptation trade-offs, which we address in Section 4.

## C  STATE REPRESENTATIONS

Following Yang et al. (2025), the state of the cloud system at decision step $t$ is represented by two graph representations: *task-specific graph* $\mathcal{G}^a(s_t, a)$ and *system-oriented graph* $\mathcal{G}^c(s_t)$, tailored to the roles of the actor and critic networks, respectively.

**Task-specific graph** $\mathcal{G}^a(s_t, a)$ is presented to each **Actor** in the ensemble, as shown in Figure 6. This graph explicitly models the consequence of assigning the focused task $O_t^*$ to a specific machine $M_q$ by updating node features and inserting a topological connection to that machine's last task, enabling a nuanced comparison between candidate actions.

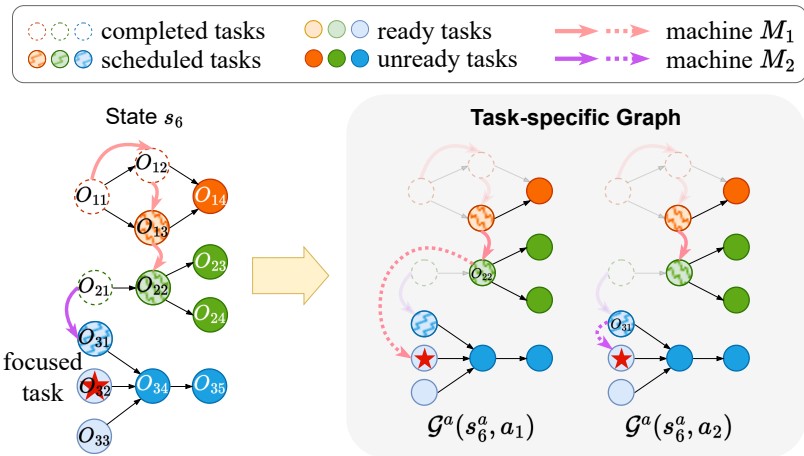

Figure 6: **Task-specific graph representation for actor networks.**

The actor network takes as input a pair $(s_t^a, a_t)$, represented by the task-specific graph $\mathcal{G}^a(s_t^a, a_t) = (\mathcal{O} \setminus \mathcal{O}_t^e, \mathcal{C}_\mathcal{W} \cup \mathcal{C}_\mathcal{M} \cup \mathcal{C}_{a_t})$. Here, completed tasks $\mathcal{O}_t^e$ are excluded since their execution is irreversible, which reduces the graph size and improves efficiency. The arc set $\mathcal{C}_\mathcal{W} = \{\mathcal{C}_{W_i} | W_i \in \mathcal{G}^a(s_t^a, a_t)\}$ encodes precedence constraints among unfinished tasks, while $\mathcal{C}_\mathcal{M} = \{\mathcal{C}_{M_q} | M_q \in \mathcal{G}^a(s_t^a, a_t)\}$ models the task-ordering within each machine's queue. To incorporate the action $a_t$, an additional arc set $\mathcal{C}_{a_t} = \{(O_{last}^{(a_t)}, O_t^*)\}$ is introduced, connecting the last pending task on machine $M_{a_t}$ to the focused task $O_t^*$. If $M_{a_t}$ has no pending tasks, a self-loop $(O_t^*, O_t^*)$ is used. This design explicitly encodes the scheduling decision within the actor's input graph.

**System-oriented graph** $\mathcal{G}^c(s_t)$ is presented to each **Critic** in ensemble, as shown in Figure 7. This graph holistically encodes the entire system state, including all uncompleted tasks, workflow dependencies, machine statuses, and queueing constraints, providing a comprehensive basis for estimating the long-term value of a state.

The critic network operates on a broader system view represented by $\mathcal{G}^c(s_t^c) = (\mathcal{O} \setminus \mathcal{O}_t^e, \mathcal{C}_\mathcal{W} \cup \mathcal{C}_\mathcal{M} \cup \mathcal{C}_\mathcal{A})$. In addition to precedence and queue arcs, a virtual arc set $\mathcal{C}_\mathcal{A} = \{(O_{last}^{(a_t)}, O_t^*) \mid a_t \in \mathcal{A}\}$ is introduced. Each arc links the last pending task of a machine eligible to process the focused task to the node of $O_t^*$. This allows the critic to explicitly consider the relationship between every candidate machine and the focused task. By incorporating $\mathcal{C}_\mathcal{A}$, the critic graph not only provides a global system representation but also emphasizes the central role of the focused task in each scheduling decision.

## D  ARCHITECTURE DESIGN

This appendix details the separate neural architectures used for the actor and critic. The actor and critic each use graph-based encoders that operate on the task-specific and system-oriented graphs introduced in Appendix C.

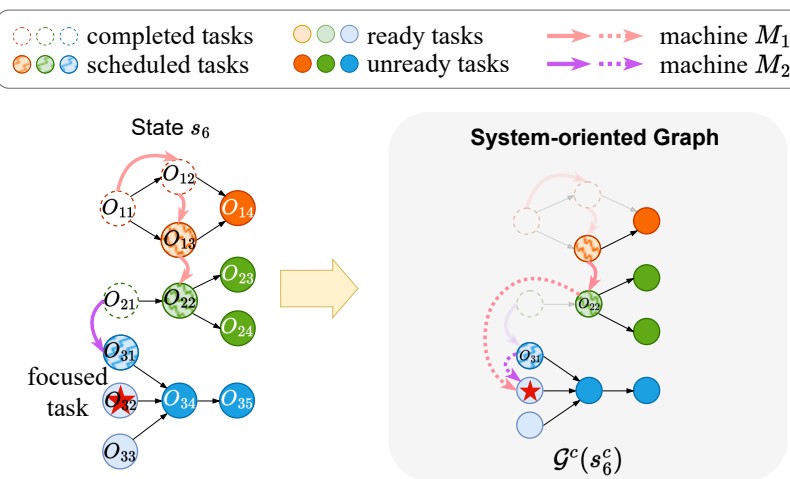

Figure 7: **System-oriented graph representation for critic networks.**

### D.1 ACTOR NETWORK ARCHITECTURE

**Task-specific embedding.** For action evaluation, the actor first encodes the task-focused observation–action pair $(s_t^a, a_t)$ using a graph neural encoder. The encoder applies $K$ layers of Graph Attention Networks (GAT) (Veličković et al., 2018) to $\mathcal{G}^a(s_t^a, a_t)$ and produces a vector representation for the focused task node $O_t^*$.

Let $\mathbf{h}_x^{(0)} = \mathbf{f}_x \in \mathbb{R}^7$ denote the input feature vector of node $x$. At layer $k \in \{1, \dots, K\}$ the node update is written as

$$\mathbf{h}_x^{(k)} = \sigma\Big( \sum_{y \in \mathcal{N}(x)} \alpha_{xy}^{(k)} \mathbf{W}^{(k)} \mathbf{h}_y^{(k-1)} \Big), \tag{17}$$

where $\mathcal{N}(x)$ is the set of neighbors of $x$ in the local graph (including both incoming and outgoing neighbors defined by $\mathcal{C}_{\mathcal{W}}$, $\mathcal{C}_{\mathcal{M}}$ and $\mathcal{C}_{a_t}$), $\alpha_{xy}^{(k)}$ are attention weights computed by the layer, $\mathbf{W}^{(k)} \in \mathbb{R}^{d \times d}$ is a learnable projection, and $\sigma(\cdot)$ denotes a nonlinearity (e.g., ReLU). The actor then extracts the final focused-task embedding

$$\hat{\mathbf{h}}_{s_t^a, a_t} = \mathbf{h}_{O_t^*}^{(K)} \in \mathbb{R}^d, \tag{18}$$

which is forwarded to the actor's action-scoring head.

**Action scoring and selection.** Given $\hat{\mathbf{h}}_{s_t^a, a_t}$, the actor computes a scalar compatibility score for the candidate action via a Multi-Layer Perceptron (MLP) layer:

$$z(s_t^a, a_t) = \mathrm{MLP}_\theta\big(\hat{\mathbf{h}}_{s_t^a, a_t}\big). \tag{19}$$

Scores for all eligible actions are collected and converted to a categorical distribution by *softmax*:

$$\pi_\theta(a \mid s_t^a) = \frac{\exp\big(z(s_t^a, a)\big)}{\sum_{a'} \exp\big(z(s_t^a, a')\big)}. \tag{20}$$

Actions are then obtained by the highest-probability action $\arg\max_a \pi_\theta(\cdot \mid s_t^a)$.

## D.2 CRITIC NETWORK ARCHITECTURE

**System-oriented embedding.** The critic uses a separate encoder to build a global representation of the system state $\mathcal{G}^c(s_t)$. This encoder first applies $K$ layers of GAT networks, analogous to the actor's local encoder, to obtain node-level features $\mathbf{e}_x^{(K)}$ for every unfinished task $x$. The per-layer update follows the same attention-based message aggregation:

$$\mathbf{e}_x^{(k)} = \sigma\Big( \sum_{y \in \mathcal{N}(x)} \tilde{\alpha}_{xy}^{(k)} \mathbf{W}_\phi^{(k)} \mathbf{e}_y^{(k-1)} \Big), \tag{21}$$

with $\tilde{\alpha}_{xy}^{(k)}$, $\mathbf{W}_\phi^{(k)}$ and $\mathcal{N}(x)$ defined analogously on the system graph (including $\mathcal{C}_\mathcal{W}$, $\mathcal{C}_\mathcal{M}$ and $\mathcal{C}_\mathcal{A}$).

To capture higher-order, global interactions among all nodes, the critic then processes the matrix of node embeddings $\mathbf{U}^{(0)} = [\mathbf{e}_x^{(K)}]_{x \in \mathcal{O}_t} \in \mathbb{R}^{|\mathcal{O}_t| \times d}$ through $L$ self-attention layers. Each self-attention layer performs a scaled dot-product attention update:

$$\mathbf{U}^{(\ell)} = \text{softmax}\Big( \frac{(\mathbf{U}^{(\ell-1)}\mathbf{W}_Q^{(\ell)})(\mathbf{U}^{(\ell-1)}\mathbf{W}_K^{(\ell)})^\top}{\sqrt{d}} \Big) \mathbf{U}^{(\ell-1)}\mathbf{W}_V^{(\ell)}, \tag{22}$$

where $\mathbf{W}_Q^{(\ell)}, \mathbf{W}_K^{(\ell)}, \mathbf{W}_V^{(\ell)} \in \mathbb{R}^{d \times d}$ are learnable matrices and $\ell \in \{1, \ldots, L\}$. The output $\mathbf{U}^{(L)}$ yields refined node embeddings $\mathbf{e}_x^{(K+L)}$ that incorporate global context.

A pooled global representation is obtained by averaging node embeddings:

$$\bar{\mathbf{e}}_{s_t^c} = \frac{1}{|\mathcal{O}_t|} \sum_{x \in \mathcal{O}_t} \mathbf{e}_x^{(K+L)}. \tag{23}$$

This pooled vector summarizes the current unfinished-task population and their interrelations.

**State value.** The critic maps the pooled global vector to a scalar state-value estimate through a MLP layer:

$$V_\phi(s_t^c) = \text{MLP}_\phi\big(\bar{\mathbf{e}}_{s_t^c}\big). \tag{24}$$

This output approximates the expected return from state $s_t^c$ under the current policy ensemble and is used in advantage computation and temporal-difference training of the critic.

## E  RAW NODE FEATURES IN GRAPH REPRESENTATION

The graph encoders operate on node-level raw features that summarize task- and machine-related information. Below we list the seven scalar features used for each task node $O_{ij}$. For machine-related features $(et, ct, ms, mu)$, the notation $^{(q)}$ denotes the value computed with respect to a specific machine $M_q$. If a node is not explicitly associated with a concrete machine, an estimated average value across machines is used and denoted by $^{(\bar{q})}$.

**Workflow-related features:**

- $es_{ij}$ — *execution status*. Encodes the current scheduling state of task $O_{ij}$ using an integer code (unassigned / focused / assigned / finished). Values are drawn from $\{0, 1, 2, 3\}$.
- $tw_{ij}$ — *task workload*. The processing demand of task $O_{ij}$ (i.e., required CPU-seconds). This scalar captures task size used for runtime estimates. The Upper bound in our settings is up to $\sim 3.9 \times 10^4$ seconds.
- $rw_{ij}$ — *remaining workflow workload*. The sum of workloads of all unfinished tasks in the same workflow $W_i$ that contains $O_{ij}$. This feature reflects the remaining work of the workflow to which the node belongs.

**Machine-related features:**

- $et_{ij}^{(q)}$ or $et_{ij}^{(\bar{q})}$ — *execution time on machine*. Estimated runtime of task $O_{ij}$ when executed on machine $M_q$ (or on the average machine $M_{\bar{q}}$ if $M_q$ is not applicable). Values depend on task size and machine speed.

- $ct_{ij}^{(q)}$ or $ct_{ij}^{(\bar{q})}$ — *expected completion time*. The projected completion time of $O_{ij}$ on machine $M_q$ (or on an average machine $M_{\bar{q}}$), accounting for the machine's current queue. This timestamp-like scalar gives a short-term scheduling horizon for the node.

- $ms_q$ or $ms_{\bar{q}}$ — *machine speed*. The processing capacity of machine $M_q$. When a node is not tied to a specific machine, the system-average speed $ms_{\bar{q}}$ is used. In our setup, machine-speed values are drawn from a small discrete set (e.g., $\{8, 16, 32, 48, 64, 96\}$).

- $mu_q$ or $mu_{\bar{q}}$ — *machine utilization*. The fraction of elapsed system time that $M_q$ has been busy (i.e., its busy-time ratio). This feature is bounded in $[0, 1)$.

# F  THE RAISE ALGORITHM

For online learning in dynamic workflow scheduling, we provide the detailed pseudo code of our proposed RAISE framework in Algorithm 1. Unlike prior DWS work, RAISE incorporates ensemble actors and dual critic ensembles to improve robustness and adaptability under *non-stationary workloads*. The algorithm alternates between online scheduling (*line 5–13*) and neural network updates (*line 14–23*).

The action aggregation follows a two-stage mechanism. At each decision step, all actors independently propose the best action for the focused task, followed by a critic-guided ranking to select the final action. This design stabilizes decision-making when actors disagree, while balancing adaptivity and conservativeness via random switching between critic ensembles.

The executed action and resulting transition are stored in a global buffer $\mathcal{B}$, and the responsible actor is also recorded in $\mathcal{I}$. This ensures that each actor is updated only on transitions aligned with its own decisions, thereby preserving policy diversity and preventing ensemble collapse. Every $T_w$ steps, after an initial warm-up ($T_{nw}$), the most recent transitions are used to update critics and actors.

# G  SIMULATION SETTINGS

## G.1  WORKFLOW PATTERNS

Our experiments adopt the common benchmark workflows used in the cloud scheduling literature and derived from the Pegasus workflow repository[3], which is a standard source of real-world scientific workflow patterns. We use four canonical workflow templates that are widely employed in prior DWS studies (Deelman et al., 2015; Huang et al., 2022; Xu et al., 2023): *Montage*, *CyberShake*, *SIPHT* and *Inspiral*. Each template specifies a DAG of tasks together with task workloads $tw_{ij}$ and precedence edges; these task-level attributes and dependencies constitute the scheduling instances used in our simulations (detailed workflow graphs are illustrated in Figure 8).

In our experiments, workflows arrive dynamically following a Poisson process, consistent with the setup in prior work (Wang et al., 2019; Sun et al., 2024; Shen et al., 2024). Table 5 summarizes basic statistics for the four template types used in this paper (number of tasks, number of precedence edges, average per-task workload, and total workflow workload).

## G.2  MACHINE CONFIGURATIONS

The simulator models a heterogeneous pool of virtual machines following common Amazon EC2[4] instance characteristics. Each machine type is parameterized by its logical vCPU count (used as the processing-speed proxy $ms_q$), memory size, and an on-demand hourly price. The execution time of a

---

[3]https://github.com/pegasus-isi/pegasus
[4]https://aws.amazon.com/ec2/pricing/on-demand/

---

**Algorithm 1: RAISE**: Ensemble-based Online RL for DWS

---

**Input:** Parameter vectors $\{\theta_i\}_{i=1}^n$ and $\{\psi_k\}_{k=1}^m$, environment configurations $args$, Polyak $\tau$, window size $T_w$, env warm-up steps $T_{nw}$, critic warm-up steps $T_{cw}$, update threshold $U_a$

**Output:** Online scheduling decisions

1 Initialize actor networks $\{\pi_{\theta_i}\}_{i=1}^n$, and conservative critic networks $\{\hat{Q}_{\psi_k}\}_{k=1}^m$;
2 Initialize adaptive critic networks $\{Q_{\phi_j}\}_{j=1}^m$ with $\forall j, \phi_j \leftarrow \psi_j$;
3 Initialize online environment $env(args)$ and an experience buffer $\mathcal{B}$ with fixed size $T_w$;
4 **for** *each decision step $t$* **do**
    // Online Decision-making
5   Observe $s_t$;
6   Each actor selects $a_{t,i}^* \leftarrow \arg\max_{a \in \mathcal{A}} \pi_{\theta_i}(a|s_t)$ to form tie set $A_t^{\text{tie}}$;
7   **if** $|A_t^{tie}| = 1$ **then**
8     $a_t \leftarrow A_t^{\text{tie}}$
9   **else**
10     Randomly pick a critic ensemble (adaptive or conservative) and rank candidates in $A_t^{\text{tie}}$;
11     $a_t \leftarrow \arg\min_{a \in A_t^{\text{tie}}} \text{AvgRank}(a, \{Q_{\phi_j}\}\text{or}\{\hat{Q}_{\psi_k}\})$ by equation 3 or equation 4;
12   Execute $a_t$, observe $r_t, s_{t+1}$, and store $(s_t, a_t, r_t, s_{t+1})$ in buffer $\mathcal{B}$;
13   Record responsible actors: $\mathcal{I}_t \leftarrow \{i : \pi_{\theta_i}(a_t|s_t) = \max_a \pi_{\theta_i}(a|s_t)\}$;
    // Periodic Updates
14   **if** $t \geq T_{nw}$ *and* $(t - T_{nw}) \bmod T_w = 0$ **then**
15     Compute returns $R$ from $\mathcal{B}$;
16     **Conservative critics** $\{\hat{Q}_{\psi_k}\}_{k=1}^m$: update by Polyak averaging using equation 7;
17     **Adaptive critics:** update by MSE to $R$ using equation 6;
18     **if** $\frac{t-T_{nw}}{T_w} \geq T_{cw}$ **then**
19       **for** *each actor $\pi_{\theta_i}$* **do**
20         Select transitions $\hat{\mathcal{B}} \leftarrow \{(s_i, a_i, r_i, s_{i+1})\}$ where $i \in \mathcal{I}$;
21         **if** $|\hat{\mathcal{B}}| > U_a$ **then**
22           Compute advantages using adaptive critics $\{Q_{\phi_j}\}$;
23           PPO update $\pi_{\theta_i}$ with gradient control (using past mean/std) using equation 8 and equation 9;
24   $s_t \leftarrow s_{t+1}$;

---

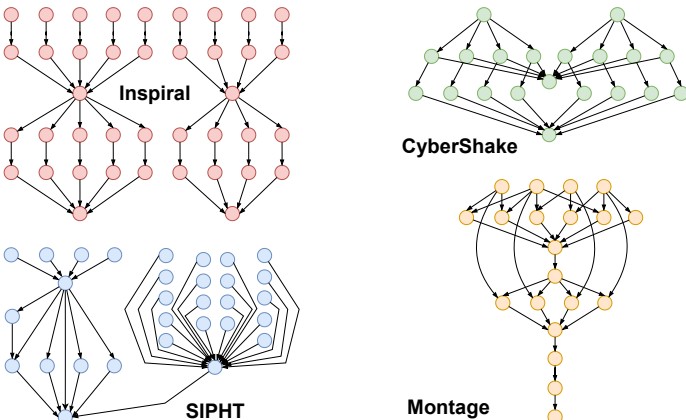

Figure 8: **Four widely used workflow patterns in cloud computing.**

task $O_{ij}$ on machine $M_q$ is computed as

$$et_{ij}^{(q)} = \frac{tw_{ij}}{ms_q},$$

Table 5: **Summary statistics of the workflow templates used in experiments.** "total workload" reports the sum of all task workloads within a workflow.

| Workflow Pattern | # tasks | # precedence edges | avg. task workload (s) | total workload (h) |
|---|---|---|---|---|
| Montage | 25 | 45 | 145.76 | 1.01 |
| CyberShake | 30 | 52 | 405.62 | 3.38 |
| SIPHT | 29 | 33 | 3060.12 | 24.65 |
| Inspiral | 30 | 35 | 3529.10 | 29.41 |

Table 6: **Machine types used in our experiments.** "vCPU" is treated as the processing-speed proxy $ms_q$. "Price": hourly rates are illustrative and rounded.

| VM type | vCPU ($ms_q$) | Memory (GiB) | Price (USD/h) |
|---|---|---|---|
| c5.large (T1) | 2 | 4 | $0.085 |
| m5.xlarge (T2) | 4 | 16 | $0.192 |
| m5.2xlarge (T3) | 8 | 32 | $0.384 |
| m5.4xlarge (T4) | 16 | 64 | $0.768 |
| m5.8xlarge (T5) | 32 | 128 | $1.536 |
| m5.16xlarge (T6) | 64 | 256 | $3.072 |

where $tw_{ij}$ is the task workload (in seconds) and $ms_q$ is the vCPU-based speed for $M_q$. For example, a task with $tw_{ij} = 402.3$ seconds will have $et_{ij}^{(q)} = 402.3/8 \approx 50.29$ seconds on a machine with $ms_q = 8$ vCPUs.

The number of machines of each type is determined by the experimental scenario. For instance, the scenario notation $\langle 4 \times 10, 5.4, 20k \rangle$ means the first four VM types (i.e., T1, T2, T3, T4) in Table 6 are used and there are ten identical machines of each type (i.e., total of $4 \times 10 = 40$ machines).

## H IMPLEMENTATION OF BASELINES

We provide additional details on the baseline methods compared with **RAISE**. The baselines span three categories: expert-designed PDRs, automatically evolved tree-based PDRs, and neural PDRs.

**Expert-designed PDRs.**

- *Earliest Start Time* (EST): A hand-crafted PDR that selects the machine capable of starting the focused task at the earliest possible time (Chetto & Chetto, 1989).
- *Predict Earliest Finish Time* (PEFT): A heuristic PDR that considers both precedence constraints and heterogeneous machine speeds to minimize the expected finish time (Senapati et al., 2021).
- *Heterogeneous Earliest Finish Time* (HEFT): A widely used heuristic for heterogeneous computing platforms, which assigns the task to the machine that can complete it earliest (Topcuoglu et al., 2002).

**Tree-based PDRs.**

- *Genetic Programming Hyper-Heuristic* (GPHH): A state-of-the-art algorithm that evolves tree-based PDRs through genetic programming. Following prior work (Xu et al., 2023; Sun et al., 2024), we adopt standard parameter settings, run 30 independent trials with different random seeds, and report the best results. The detailed parameter settings are summarized in Table 7.

**Neural PDRs.**

- *ERL-DWS*: An Evolution Strategies-based RL framework with Transformer-based feature extractors for learning neural PDRs (Shen et al., 2024). We follow the official implementations and report their best results among five different random seeds.
- *GOODRL*: An offline-online RL method that integrates graph-based actor and critic networks with imitation learning and offline-online PPO, designed for DWS (Yang et al., 2025).

For both methods, we follow the official implementations and report their best results among five different random seeds.

Table 7: **Parameter settings of GPHH baseline.**

| Parameter | Value |
|---|---|
| Population size | 1024 |
| Generations | 50 |
| Crossover rate | 0.80 |
| Mutation rate | 0.15 |
| Reproduction rate | 0.05 |
| Tournament size | 7 |
| the number of elites | 10 |
| Maximum tree depth during evolution | 8 |
| Initial depth of trees | 2 to 6 |
| Independent runs | 30 |

# I   HYPERPARAMETERS

**Network Architecture.** Each actor network $\pi_{\theta_i}$ consists of 2 Graph Attention Network (GAT) layers with one attention head, followed by 4 fully connected layers (MLPs). Each critic network (both adaptive $Q_{\phi_j}$ and conservative $\hat{Q}_{\psi_k}$) has 2 GAT layers with one attention head, 1 self-attention layer with two heads, and 4 MLP layers. All hidden layers have 128 units. To reduce computational overhead, all networks share the same lightweight architecture, enabling efficient parallel inference during ensemble decision-making.

**Normalization.** All raw input features are normalized by dividing by pre-defined constants to ensure stable training. Rewards are normalized by 1000 to maintain a consistent scale across tasks.

**Ensemble Initialization.** RAISE maintains an ensemble of $n = 5$ actors, $m = 5$ adaptive critics, and $m = 5$ conservative critics. All networks are initialized with pre-trained GOODRL models from the offline scenario $\langle 5 \times 5,\ 5.4,\ 30 \rangle$ before online adaptation. Adaptive and conservative critics are identical at initialization, after which their updates diverge following the dual critic ensemble mechanism.

**Online Learning.** RAISE trains purely online using PPO-style updates. Each training cycle alternates between (i) real-time scheduling for a fixed number of steps and (ii) updating actors and critics with transitions collected in the recent time window. Adaptive critics are updated with a high update-to-data (UTD) ratio, while conservative critics are updated slowly via Polyak averaging. Actors are updated only with transitions aligned to their past decisions, using PPO with gradient clipping to stabilize training.

Table 8: **RAISE hyperparameters.**

| Hyperparameter | Value |
|---|---|
| Actors $(n)$ | 5 |
| Adaptive critics $(m)$ | 5 |
| Conservative critics $(m)$ | 5 |
| Critic warm-up windows | 50 |
| Window size $(T_w)$ | 2048 |
| Update threshold $(U_a)$ | 512 |
| Mini-batch size | 64 |
| Update-to-data ratio (actors) | 4 |
| Update-to-data ratio (critics) | 16 |
| Clipping parameter $(\epsilon)$ | 0.2 |
| Discount factor $(\gamma)$ | 0.99 |
| Actor learning rate | $5 \times 10^{-5}$ |
| Critic learning rate | $1 \times 10^{-4}$ |
| Polyak factor $(\tau)$ | 0.005 |
| Max gradient norm $(\tau_0)$ | 0.075 |

**Hardware and Implementation.** We implement GAT-based networks using PyTorch-Geometric (Fey & Lenssen, 2019), and other modules with PyTorch (Paszke et al., 2019). Experiments are conducted in a cloud computing environment dominated by CPU resources. Each experiment is run on 1 compute node, each with 1 CPU and 10 GB of memory per CPU.

Table 9: **Software and hardware version information.**

| Software | Version |
|---|---|
| Python | 3.11.5 |
| PyTorch | 2.4.1 |
| PyTorch-Geometric | 2.5.3 |
| rl-zoo3 | 2.3.0 |
| deap | 1.4.1 |

## J    CRITIC SELECTION STRATEGY IN VRAA

In the Value-Ranked Action Aggregation (VRAA) mechanism (Section 4.2), RAISE randomly switches between the Adaptive Critic Ensemble and the Conservative Critic Ensemble to generate action rankings. We compared this "Random" strategy against a "Both" strategy, where rankings are derived by averaging the outputs of both critic sets simultaneously.

Table 10: **Relative improvement over single-policy in online scenario** $\langle 6 \times 4, \; 12.6, \; 20k \rangle$.

| Mechanism | 60-th | 80-th | 100-th | 120-th | 140-th | 160-th |
|---|---|---|---|---|---|---|
| **Random** | 10.16% | 12.73% | 13.13% | 13.43% | 13.39% | 13.23% |
| Both | 10.08% | 11.81% | 12.06% | 12.11% | 12.49% | 12.24% |

Table 10 reports the relative improvement (see equation 10) of both strategies over the single-policy baseline (e.g., GOODRL) across varying numbers of online interaction steps. The "Random" strategy consistently outperforms "Both" across all intervals. Furthermore, the "Both" strategy necessitates forward passes through all critic networks (adaptive and conservative critics) for every decision, significantly increasing inference overhead. Therefore, "Random" selection provides a superior trade-off between adaptation performance and computational efficiency.

## K    ACTOR SELECTION STATISTICS

In the Value-Ranked Action Aggregation (VRAA) mechanism, critics are randomly selected (Adaptive or Conservative) to rank candidate actions during tie-breaking. A potential concern is whether this random selection might lead to actor polarization, where certain actors specialize in satisfying only one type of critic while others are ignored.

To investigate this, we tracked the selection probability of each actor in the ensemble ($n = 5$) across different online interaction steps (10k to 70k). We analyzed two distinct scenarios:

- **High Variance Group** (Table 11): A set of actors where the initial performance gap among actors is relatively large.
- **Low Variance Group** (Table 12): A set of actors where actors have similar performance levels.

As shown in Table 11 and Table 12, we found no evidence of any actor being marginalized. Even when hypothetically using only Adaptive or only Conservative critics for ranking, every actor maintained a selection probability well above 15% (theoretical uniform probability is 20%). This confirms that all actors continue to contribute to decisions and receive gradient updates, preventing ensemble collapse.

The degree of selection uniformity is primarily driven by the performance variance among actors rather than the critic type. In the High Variance Group, selection probabilities ranged from 15.93% to 22.81%. In the Low Variance Group, probabilities were more uniform, ranging from 18.26% to 21.69%. Therefore, the random critic selection strategy in RAISE does not induce update polarization.

Table 11: **Actor selection statistics on a high variance group of actors.**

| Type | Env steps | Actor-1 | Actor-2 | Actor-3 | Actor-4 | Actor-5 | Sum |
|------|-----------|---------|---------|---------|---------|---------|-----|
| Adaptive | 10k | 20.59% | 16.21% | 20.66% | 20.10% | 22.44% | 100% |
|  | 30k | 17.68% | 18.34% | 20.99% | 20.53% | 22.46% | 100% |
|  | 50k | 17.15% | 18.83% | 20.91% | 20.59% | 22.52% | 100% |
|  | 70k | 16.83% | 19.33% | 21.13% | 19.90% | 22.81% | 100% |
| Conservative | 10k | 20.52% | 15.93% | 20.97% | 20.13% | 22.45% | 100% |
|  | 30k | 17.68% | 17.79% | 21.31% | 20.65% | 22.57% | 100% |
|  | 50k | 17.37% | 18.10% | 21.06% | 20.96% | 22.51% | 100% |
|  | 70k | 17.27% | 18.49% | 21.13% | 20.41% | 22.70% | 100% |

Table 12: **Actor selection statistics on a low variance group of actors.**

| Type | Env steps | Actor-1 | Actor-2 | Actor-3 | Actor-4 | Actor-5 | Sum |
|------|-----------|---------|---------|---------|---------|---------|-----|
| Adaptive | 10k | 20.36% | 21.33% | 19.21% | 18.26% | 20.84% | 100% |
|  | 30k | 20.73% | 21.69% | 19.87% | 18.52% | 19.18% | 100% |
|  | 50k | 20.99% | 20.78% | 20.55% | 19.31% | 18.37% | 100% |
|  | 70k | 21.26% | 19.87% | 20.94% | 19.59% | 18.35% | 100% |
| Conservative | 10k | 20.32% | 20.93% | 19.63% | 18.57% | 20.54% | 100% |
|  | 30k | 20.23% | 21.43% | 20.23% | 18.95% | 19.16% | 100% |
|  | 50k | 20.45% | 20.62% | 20.76% | 19.72% | 18.44% | 100% |
|  | 70k | 20.73% | 19.86% | 21.04% | 19.88% | 18.49% | 100% |

## L  ENERGY EFFICIENCY ANALYSIS

To assess the real-world practicality of RAISE, we estimated the energy consumption of its inference process using the Thermal Design Power (TDP) of standard cloud hardware.

**Inference on CPU.** The experiments reported in Table 1 were conducted on an **AMD EPYC 7702 64-Core Processor**, which has a default TDP of **200W**[5]. Based on the measured inference times ranging from 0.0341s to 0.0864s per decision, the estimated energy consumption is calculated as:

$$E_{cpu} \approx P_{cpu} \times t_{inf} \approx 200\text{W} \times (0.0341\text{s} \sim 0.0864\text{s}) \approx \mathbf{6.82J - 17.28J} \tag{25}$$

**Inference on GPU.** Deployment on GPUs can accelerate inference. On an **NVIDIA A100-PCIE-40GB** with TDP **250W**[6], our model inference achieves a speedup of approximately $4\times$ compared to CPUs. This reduces the estimated inference time to $\sim 0.0085\text{s} - 0.0216\text{s}$ per decision, yielding the following energy estimate:

$$E_{gpu} \approx P_{gpu} \times t_{inf} \approx 250\text{W} \times (0.0085\text{s} \sim 0.0216\text{s}) \approx \mathbf{2.13J - 5.40J} \tag{26}$$

These energy costs are negligible compared to the total energy consumed by the cloud resources executing the actual workflows. For example, a single workflow task running for just 1 minute on a similar machine consumes approximately **12,000J** ($60\text{s} \times 200\text{W}$). Consequently, the energy overhead of RAISE's decision-making is orders of magnitude lower ($< 0.1\%$) than the execution energy it manages. Given that RAISE reduces the total mean-flowtime (and thus server active time) by over 4% in many scenarios, the net energy savings far outweigh the inference costs.

## M  STATISTICAL ANALYSIS OF DUAL CRITIC ENSEMBLES

In Section 4.2, we visualized the distinct behaviors of the Adaptive and Conservative critic ensembles using heatmaps (Figure 2). To provide a quantitative basis for these observations, we recorded the

---

[5]https://www.amd.com/en/products/processors/server/epyc/7002-series.html
[6]https://www.nvidia.com/content/dam/en-zz/Solutions/Data-Center/a100/pdf/A100-PCIE-Prduct-Brief.pdf

Table 13: **Q-value distributions on the same set of states and actions.**

| Critics | Mean | Std. | Range |
|---|---|---|---|
| Conservative Critics | -0.8367 | 0.2821 | [-1.7795, -0.0683] |
| Adaptive Critics | -3.3997 | 1.6314 | [-6.7356, -0.9814] |

Q-value estimates from both ensembles on the same set of state-action pairs sampled during online scheduling. Table 13 summarizes the statistical properties of these distributions.

**Stability (Standard Deviation).** The standard deviation of the **Adaptive Critics** (1.6314) is higher than that of the **Conservative Critics** (0.2821). This quantitative gap confirms that the Adaptive ensemble is highly sensitive to recent environmental fluctuations and state changes, whereas the Conservative ensemble acts as a stable anchor, resisting rapid variance.

**Accuracy (Value Range).** There is a significant discrepancy in the value ranges. Since the reward function is defined as negative flowtime, lower values indicate higher congestion costs. The **Adaptive Critics** cover a much wider and lower range ([-6.7356, -0.9814]), correctly reflecting the severe costs of the current online environment. In contrast, the **Conservative Critics** yield estimates closer to zero (Mean $-0.8367$), indicating they are biased towards the offline data distribution.

## N  SAMPLE EFFICIENCY ANALYSIS

To rigorously evaluate sample efficiency, we tracked the **Relative Improvement** of RAISE and GOODRL compared to the heuristic baseline (i.e., HEFT) across strictly aligned online interaction steps (number of samples). Table 14 presents the performance progression. The empirical findings confirm that RAISE improves sample efficiency significantly compared to the single-policy baseline:

- **Rapid Adaptation:** RAISE achieves immediate positive gains. With only **147k samples**, RAISE already outperforms HEFT by **7.99%**, whereas GOODRL performs worse than the heuristic ($-2.41\%$) at the same stage. This indicates that RAISE requires significantly less "warm-up" data to learn effective policies in online environments.
- **Superior Growth:** As training progresses to **345k samples**, RAISE's improvement reaches **23.42%**, maintaining an approximately **2×** performance lead over GOODRL (11.74%). This demonstrates that the ensemble-based update mechanism extracts value from streaming data more effectively than single-policy approaches.

Table 14: **Relative performance improvement over HEFT across online interaction steps.**

| | 147k | 187k | 226k | 266k | 305k | 345k |
|---|---|---|---|---|---|---|
| GOODRL | $-2.41\%$ | $-0.16\%$ | $4.44\%$ | $7.79\%$ | $10.25\%$ | $11.74\%$ |
| RAISE | $7.99\%$ | $12.58\%$ | $16.99\%$ | $20.18\%$ | $22.26\%$ | $23.42\%$ |

## O  PRACTICAL IMPLICATIONS

While a quantitative improvement of approximately 4% in mean-flowtime **compared to learning-based baselines** might appear marginal in isolation, we emphasize that in the context of large-scale cloud scheduling, this represents a highly meaningful breakthrough.

**Algorithmic Significance.** In the mature field of distributed computing, the Heterogeneous Earliest Finish Time (HEFT) algorithm (Topcuoglu et al., 2002) remains a foundational heuristic with over 4,600 citations, having been optimized and validated for decades. Against such a robust baseline, RAISE achieves a performance improvement of up to **29.61%**. In operations research, consistently outperforming a mature, industry-standard heuristic by a double-digit margin represents a substantial algorithmic advancement rather than a marginal gain.

**Operational Scale.** Major cloud providers (e.g., AWS, Azure, GCP) manage billions of processes daily. Consider a hyperscale data center processing approximately **500,000 workflows per day**. In this context, a **4% reduction** in mean-flowtime translates to **millions of operational seconds** saved every single day. This reduction in end-to-end delay directly enhances system throughput,

reduces the probability of Service Level Agreement (SLA) violations, and significantly improves user responsiveness at scale.

**Economic and Environmental Impact.** Efficiency in workflow scheduling directly correlates with the "active time" of high-power servers. By reducing the mean-flowtime, RAISE allows computing resources to return to idle or low-power states sooner. Applied to a standard large-scale cluster, a 4% reduction in active duty cycles can lower energy consumption by **thousands of kWh per day**. Over long-term operations, this yields massive cost savings and contributes to a measurable reduction in the data center's carbon footprint, aligning with the growing industry priority on sustainable computing.

## P    RATIONALE AND ANALYSIS OF GRADIENT DROPPING

To ensure stable learning under highly non-stationary online conditions, RAISE employs a **Double Protection Strategy** for policy updates.

**Mechanism Rationale.** First, we adopt the standard PPO clipping mechanism, which constrains the policy update ratio $r_t(\theta)$ to prevent destructive large updates. However, ratio clipping alone does not directly regulate the absolute scale of policy gradients. In dynamic DWS environments, outlier batches or abrupt distributional shifts can produce unusually large gradient norms even if the probability ratio is clipped. These high-norm gradients can propagate through the actor network, causing instability.

To address this, the gradient-dropping rule (equation 9) provides a second, more stringent layer of protection. By discarding updates whose gradient magnitudes exceed adaptive statistical thresholds ($\mu_{\text{prev}} + \sigma_{\text{prev}}$), this mechanism explicitly filters out unstable or noisy batches. This conservative safeguard is critical for preventing "bad" updates from propagating through the ensemble and causing long-term performance degradation in sequential decision-making.

**Frequency Analysis.** To verify that this mechanism acts as a safety filter rather than a hindrance to learning, we tracked the cumulative frequency of dropped gradients across different online interaction steps. Table 15 presents the statistics for two representative scenarios. The data confirms that the mechanism effectively filters out high-variance noise throughout the lifelong learning process.

Table 15: **Cumulative frequency of activating gradient control across online interaction steps.**

|  | 147k | 187k | 226k | 266k | 305k | 345k |
|---|---|---|---|---|---|---|
| $\langle 6 \times 4,\ 9.0,\ 20k \rangle$ | 771 | 2419 | 3997 | 5636 | 7286 | 8932 |
| $\langle 5 \times 7,\ 5.4,\ 20k \rangle$ | 615 | 2196 | 3906 | 5500 | 7011 | 8721 |

## Q    SCALABILITY AND COMPLEXITY ANALYSIS

To determine the optimal ensemble size and justify the complexity-performance trade-off, we conducted a scalability analysis with ensemble sizes $n \in \{3, 5, 10, 15\}$. We evaluated the impact of $n$ on the scheduling objective (Mean-Flowtime), stability (Standard Deviation across seeds), and computational cost (Average Inference Time on CPU).

As summarized in Table 16, the results support our design choice:

- **Performance vs. Stability:** Increasing $n$ from 3 to 5 yields a significant performance gain (Objective reduces from 307.21 to 301.78) and a substantial improvement in stability (Std reduces from 5.01 to 3.02).
- **Diminishing Returns:** Further increasing $n$ to 10 or 15 yields only marginal gains in the objective ($< 1.5\%$ improvement) while increasing the computational footprint.
- **Optimal Balance:** We selected $n = 5$ as the "sweet spot." It achieves near-optimal performance and high stability with a negligible inference overhead of $\approx 0.05s$ (on CPU), which is orders of magnitude lower than the execution time of the workflows it schedules.

Table 16: **Scalability analysis across different ensemble sizes.** *Obj.* denotes mean-flowtime. Inference time is measured on an AMD EPYC 7702 Processor.

| Ensemble Size | Objective | Std. (Stability) | Avg. Inference Time (s) |
|---|---|---|---|
| $n = 3$ | 307.21 | 5.01 | 0.0233 |
| $n = 5$ **(Ours)** | **301.78** | **3.02** | **0.0494** |
| $n = 10$ | 297.77 | — | 0.1024 |
| $n = 15$ | 297.49 | — | 0.0698 |

