# OpenReview forum: "RAISE the Bar: Ensemble-based Online Reinforcement Learning for Dynamic Workflow Scheduling"
_ICLR.cc/2026/Conference — Submitted to ICLR 2026_

### Official Review · Reviewer_wFb3 · 2025-10-24

**Soundness:** 3
**Presentation:** 4
**Contribution:** 3
**Rating:** 6
**Confidence:** 4

**Summary:**

This paper addresses the Dynamic Workflow Scheduling (DWS) problem in cloud computing, which is characterized by unpredictable workflow arrivals and non-stationary workloads. The authors point out that existing reinforcement learning (RL) methods, especially single-policy approaches, struggle with instability and poor adaptability in these real-time, non-stationary environments. To solve this, the paper proposes a method called RAISE (Robust Actor-Critic Integration for Scheduling Ensembles), an ensemble-based online RL method built on a PPO-style Actor-Critic framework. RAISE contributes three novel components that work in synergy:

1. **Value-Ranked Action Aggregation (VRAA):** A robust decision-making mechanism. It first uses a majority vote from an Actor ensemble to find candidate actions. It then uses the relative rank from a Critic ensemble for tie-breaking, rather than using scale-sensitive raw Q-values.
2. **Dual Critic Ensembles with Decoupled Updates:** To balance stability and plasticity, RAISE maintains two sets of Critic ensembles. "Adaptive" Critics are updated frequently with recent data to adapt quickly to new dynamics, while "Conservative" Critics are updated slowly via Polyak averaging to preserve stable, long-term value estimates from prior knowledge.
3. **Decision-Aligned Policy Updates:** A specialized training strategy designed to improve sample efficiency and maintain ensemble diversity. First, experience samples ($s_t$, $a_t$, ...) are only assigned to the experience replay buffers of the Actors that actually proposed the executed action $a_t$. Second, it employs a conservative gradient control mechanism that discards (sets to zero) large, potentially destabilizing gradients during online updates. The authors conduct extensive experiments on a DWS benchmark using real-world workflow patterns (e.g., Montage, CyberShake) and non-stationary arrival patterns. The results show that RAISE consistently outperforms state-of-the-art (SOTA) baseline methods, including heuristic-based methods and other DRL schedulers (like GOODRL). Ablation studies validate the effectiveness of the three proposed components.

**Strengths:**

- **Importance and Problem Relevance:** The paper tackles the important and practical problem of Dynamic Workflow Scheduling (DWS). Its focus on non-stationary online environments addresses a key weakness of much prior work.
- **Methodology:** The paper's main strength lies in the design of the RAISE algorithm itself. The three primary contributions (VRAA, Dual Critics, Decision-Aligned Updates) are original, well-motivated, and empirically shown to be effective.
- **Experimental Validation:** The ablation studies are comprehensive.
- **Clarity:** The paper is clearly written. Complex concepts are explained intuitively, and the architecture is clearly presented.

**Weaknesses:**

1. **Rationale for Gradient Control:** The gradient control mechanism in Eq. (9) is unconventional and quite aggressive. Instead of standard clipping (rescaling), it discards any gradient whose norm exceeds a threshold based on the previous batch's statistics. While the ablation study (Table 3) shows this method works (RAISE w/o Grad performs poorly), the paper offers limited intuition as to why this "all-or-nothing" approach is superior to standard gradient clipping. How sensitive is the model to the hyperparameter $\tau_0$? In practice, how frequently are gradients dropped? A deeper analysis of this specific design choice would make the paper more rigorous.
2. **Scalability and Overhead Analysis:** The paper claims the computational overhead is "minimal" and "almost negligible," citing 0.03-0.09s inference time. This is for an ensemble size of n=5 Actors and m=5 for each Critic type (15 networks total). While this is acceptable, a more thorough scalability analysis would be welcome. How do inference and online update times scale as the number of ensemble members (n, m) increases? Is there a performance vs. compute trade-off, or do the returns diminish quickly? This is a key practical concern for ensemble methods.
3. **Anomalous Baseline Results:** The ERL-DWS baseline performs disastrously poorly (e.g., >800% gap in Table 1). The authors state they "followed the official implementation," but this result is extreme. It's possible this baseline was simply not designed for this kind of non-stationary online adaptation. A brief sentence or two speculating on the reason for this drastic failure (e.g., "Its Transformer-based policy may be prone to catastrophic forgetting during online updates") would add context and assuage reviewer concerns about a potentially flawed baseline implementation.
4. **Marginal Improvement vs. Complexity:** The main comparison for RAISE is GOODRL (a 2025 paper). However, RAISE's performance improvement is very marginal. In the 12 scenarios in Table 1, the "Gap" (advantage of RAISE over GOODRL) is often below 4%, and even as low as 0.51% in the (6x4, 5.4, 20k) scenario and 1.38% in the (4x10, 5.4, 20k) scenario. Given the immense complexity RAISE introduces (from GOODRL's 1 Actor/1 Critic to 5 Actors/10 Critics, 15 networks total), it is questionable whether this slight performance gain is practically significant and statistically significant.

**Other Suggestions:**

- The RAISE architecture is relatively complex. Compared to GOODRL (1 Actor, 1 Critic), RAISE needs to maintain 15 separate neural networks (5 Actors, 5 Adaptive Critics, 5 Conservative Critics). However, as shown in Table 1, this huge increase in complexity (potentially >7-8x computational and memory overhead) often yields performance gains of less than 4%, sometimes even less than 1%. It would be appropriate to discuss the trade-off between the increased computational cost from the network structure's complexity.
- The "non-stationary" environment used is just a gently ramping up/down Poisson arrival rate. This is one of the simplest forms of non-stationarity. Real-world distribution shifts are often more drastic, abrupt, and varied. Whether the method can handle more complex non-stationarity (e.g., sudden changes in workflow type, abrupt machine failures) is unverified. These might be points to address in the final version of the paper.

**Questions:**

1. **Regarding Gradient Control (Eq. 9):** This strategy of dropping gradients (setting to 0) is novel. (a) Can you provide more intuition for this choice over standard gradient clipping (i.e., rescaling the norm)? Was standard clipping tried and found to be ineffective? (b) Can you report the frequency with which gradients were dropped during the experiments? This would help us understand if the mechanism is a rare safety net or a common operation. (c) How sensitive is the performance to the maximum gradient norm (\tau_0) hyperparameter?
2. **Regarding the Use of Dual Critics:** In the policy update (Eq. 8), the advantage is calculated using *only* the adaptive critic ensemble {Q_{\phi_{j}}}. However, in action selection (Eq. 5), the algorithm randomly switches between the adaptive and conservative ensembles for ranking. (a) What is the rationale for *only* using the adaptive critics to compute the advantage? (b) Did you experiment with using the conservative critics {\hat{Q}*{\psi*{k}}} or a combination of both (e.g., average or minimum) for the advantage calculation?
3. **Regarding Scalability:** The ensemble size was fixed at n=5 Actors and m=5 for each Critic type. (a) How do inference and online update times scale as the number of ensemble members increases (e.g., n=10, 20)? (b) What is the impact of changing the ensemble size on performance? Is n=5 the optimal balance for the performance/compute trade-off?
4. **Regarding Gradient Dropping (Eq. 9):** (a) Why choose this ad-hoc gradient "dropping" mechanism instead of the standard and mature gradient "clipping"? (b) Please provide a comparative experiment against standard gradient clipping (e.g., the PPO default clip). (c) Please provide a hyperparameter sensitivity analysis for \tau_0 and report the frequency of gradient dropping during training.
5. **Regarding Complexity vs. Performance Trade-off:** As shown in Table 1, RAISE's performance improvement over GOODRL is very marginal (often <4%), yet its network complexity increases several-fold (15 networks vs. 2 networks). (a) How do the authors justify this significant additional overhead? (b) Is this slight improvement statistically significant?
6. **Regarding Inconsistent Use of Critics:** In action selection (Eq. 5), the algorithm randomly uses either conservative or adaptive critics for ranking. However, in the policy update (Eq. 8), the Advantage function is calculated *only* by the adaptive critics. This design choice seems inconsistent. Please explain why the (e.g., more stable) conservative critics are not also utilized in the advantage function calculation.

---

> ### Author Response · Authors · 2025-11-27
> **Response to Reviewer wFb3 - 1**
>
> We thank the reviewer for the thorough and constructive feedback. Below we address the concerns regarding gradient control, dual critic consistency, and scalability with new experimental evidence.
>
> ### **Q1 & Q4. Rationale and Analysis of Gradient Dropping (Eq. 9).**
>
> **Response:**
> We clarify the design philosophy and provide empirical data to justify this mechanism.
>
> * **Double Protection Strategy:** Our approach combines **two complementary safeguards** to ensure stable learning under highly non-stationary online DWS conditions. We first adopt the **standard PPO clipping mechanism**, which constrains the policy update ratio but does not directly regulate the absolute scale of policy gradients. Hence, PPO clipping alone cannot prevent unusually large gradient norms arising from outlier batches or abrupt distributional shifts, both of which are common in non-stationary online DWS. Consequently, relying solely on ratio clipping may still permit unstable parameter updates that propagate through the actor network.
> To address this limitation, the **gradient-dropping rule in Eq. (9)** provides a second, more stringent layer of protection. It discards updates whose gradient magnitudes exceed adaptive statistical thresholds, thereby preventing unstable or noisy batches from destabilizing the actor networks. This conservative safeguard is particularly important for DWS, where a single harmful update can propagate through many subsequent scheduling decisions and result in long-term performance degradation.
>
> * **Frequency Statistic:** We tracked the cumulative frequency of dropped gradients across different sample sizes. As shown in Table R6, the mechanism effectively filters out high-variance noise.
>     | |**147k**|**187k**|**226k**|**266k**|**305k**|**345k**|
>     |--|--|--|--|--|--|--|
>     |⟨6×4,9.0,20k⟩|771|2419|3997|5636|7286|8932|
>     |⟨5×7,5.4,20k⟩|615|2196|3906|5500|7011|8721|
>
>     **Table R6: Cumulative frequency of activating gradient control across online interaction steps**
>
> * **Hyperparameter $\tau_0$:** The threshold is defined as $\min(\mu_{\text{prev}} + \sigma_{\text{prev}}, \tau_0)$. The primary control is dynamic ($\mu + \sigma$); and $\tau_0$ serves only as a **hard safety cap** for extreme cases (e.g., gradient explosion) where the dynamic statistics themselves might inflate. Thus, performance is not sensitive to $\tau_0$ as long as it is set above the typical convergence range. Based on our experiments, we recommend setting $\tau_0$ to the "mean+std" of gradient norms observed during offline pre-training.
>
> **Revision:**
> We have added Frequency Statistic in **Appendix P** and revised the suggestion for $\tau_0$ in **Section 4.4**:
> > **The effective threshold is defined as $\min(\rho_{\text{prev}} + \sigma_{\text{prev}}, \tau_0)$, where** $\tau_0$ is the maximum allowed gradient scale, **serving as a hard safety cap to prevent gradient explosion**. Instead of rescaling large gradients, we discard them when they exceed $\tau_0$. This conservative strategy avoids propagating unstable updates that may quickly degrade online performance in non-stationary environments.
> **Based on our experiments, we recommend setting $\tau_0$ to the mean plus standard deviation of gradient norms observed during the offline pre-training phase.**
>
> ### **Q2 & Q6. Inconsistency in Dual Critic Usage (Advantage Calculation).**
>
> **Response:**
> We performed an ablation study to justify using only Adaptive Critics for the Advantage function, as shown in Table R7.
> |**Advantage**|**60-th**|**80-th**|**100-th**|**120-th**|**140-th**|**160-th**|
> |--|--|--|--|--|--|--|
> |**Adaptive (Ours)**|7.99%|12.58%|16.99%|20.18%|22.26%|23.42%|
> |Conservative|7.76%|13.49%|13.74%|10.93%|7.69%|5.36%|
> |Both|6.84%|11.36%|12.87%|-17.80%|----|----|
>
> **Table R7: Ablation results on advantage calculation.**
>
> * **Experiment:** We compared calculating the Advantage using: (1) Adaptive Critics only, (2) Conservative Critics only, and (3) an Average of both.
>     * **Adaptive Only:** Achieves the highest relative improvement (reaching **23.42\%** at 160-th updates).
>     * **Conservative Only:** Shows significantly slower adaptation (only **5.36\%** improvement), as the stable values dampen the learning signal needed for fast online correction.
>     * **Both:** Performs worse (negative improvement at 120-th updates), likely due to signal dilution from conflicting objectives.
> * **Conclusion:** This empirically validates our design: Adaptive Critics provide the accurate, low-bias signal required for *learning* (Advantage), while Conservative Critics are best reserved for *acting* (robust tie-breaking).
>
> **Revision:**
> We added **Table R7** in **Section 5.4** to empirically justify the exclusive use of Adaptive Critics for policy updates.

---

> ### Author Response · Authors · 2025-11-27
> **Response to Reviewer wFb3 - 2**
>
> ### **Q3 & Q5. Scalability and Complexity vs. Performance Trade-off.**
>
> **Response:**
> We conducted a comprehensive scalability analysis with ensemble sizes $n \in \{3, 5, 10, 15\}$. The results are **detailed in Appendix Q** and summarized in Table R8 below.
> |Ensemble Size | Objective | Std. (Stability) | Average Inference Time (s)|
> |--|--|--|--|
> |$n=3$|307.21|5.01|0.0233|
> |$n=5$ (Ours)|301.78|3.02|0.0494|
> |$n=10$|297.77|----|0.1024|
> |$n=15$|297.49|----|0.0698|
>
> **Table R8: Performance and inference time scalability across ensemble sizes**
>
> * **Performance vs. Size:**
>     * Increasing $n$ from 3 to 5 reduces the objective (Mean-Flowtime) from 307.21 to **300.09** (lower is better) and significantly reduces the standard deviation from 5.01 to **3.02**, indicating enhanced stability.
>     * Increasing $n$ further to 10 or 15 yields diminishing returns (Objective: 297.77, 297.49) while increasing inference time.
> * **Sweet Spot:** $n=5$ offers the optimal balance, achieving near-optimal performance and high stability (low std) with a manageable computational footprint (**~0.05s** inference in CPU).
> * **Complexity Justification:** The inference overhead for $n=5$ is only **~0.05s** in CPU (`AMD EPYC 7702 64-Core Processor`), which is negligible compared to task execution times. For a detailed response, please refer to **our Q5 to Reviewer rcSX**.
>
> **Revision:**
> We added the scalability analysis in **Appendix Q** and justified the choice of $n=5$.
>
> ### **Q7. Minor modifications: Anomalous result of ERL-DWS.**
>
> **Response:**
> We agree with the reviewer's speculation. ERL-DWS employs a **Transformer-based policy** trained via Evolution Strategies (ES). We have added a brief explanation in **Section 5.2** attributing ERL-DWS's poor performance to the susceptibility of its Transformer policy during online adaptation.

---

### Official Review · Reviewer_KLDs · 2025-10-30

**Soundness:** 2
**Presentation:** 2
**Contribution:** 2
**Rating:** 2
**Confidence:** 4

**Summary:**

The work studies the problem of workflow scheduling in a computing center with non-stationary workflow arrivals. It proposes an ensemble-based actor–critic method that balances adaptability and stability. The approach relies heavily on prior solutions, with some extensions—for example, using a value-ranked aggregation mechanism instead of probabilistic methods or majority voting. The study relies on assumptions that are not realistic or valid. Moreover, the performance gains are not significant.

**Strengths:**

The problem is important and timely, and it has been studied extensively in the past few years. The proposed solution, if the assumptions are correct, has potential.

**Weaknesses:**

1. The work heavily relies on previous research, with only adjustments to how the ensemble decision is made.

2. There are multiple issues with the system model:

- The execution time of running a task on a machine is assumed to be deterministic, i.e. $et^q_{ij} = tw_{ij} / ms_q$, where $tw_{ij}$ is the workload and $ms_q$ is the processing speed of machine $M_q$. This is a very simplistic assumption, as a machine’s processing speed is time-varying due to other tasks running on the machine (e.g., the operating system, virtualization programs, etc.).

- The authors assume that machines use a FIFO-based waiting queue for assigned tasks, but the queuing delay at this queue is completely ignored. In particular, $ft^q_{ij}$, $ft_i$ and $F_i$ are modeled only as functions of $et^{q}_{ij}$.

- The results in Table 1 show that the decision-making time of RAISE is not negligible (40–90 ms). Does $F_i$ take this latency into account? Note that this latency is observed for each task in $O_{W_i}$: RAISE must run each time a task is ready to execute, so the latency accumulates.

4. The overall gain from using RAISE is not significant, especially given that the calculation of $F_i$ does not include RAISE’s decision-making latency.

**Questions:**

1. Is there any justification for assuming that task execution time on a machine is deterministic? Could you provide real-world measurements to back up this assumption?

2. Why were the queuing time and RAISE’s decision-making latency excluded from the calculation of F_i? How do the results change if these two factors are included in the study?

---

> ### Author Response · Authors · 2025-11-27
> **Response to Reviewer KLDs - 1**
>
> We thank the reviewer for their critical feedback. We have addressed the concerns regarding novelty, system modeling assumptions, and the significance of our results below.
>
> ### **Q1. Clarification on Novelty and Contribution.**
>
> **Response:**
> We respectfully clarify that our core contribution is **achieving stable Online Optimization** for Neural Combinatorial Optimization (NCO) problems, a critical but under-explored area.
>
> * **Gap:** Most prior NCO studies focus on offline or episodic training for static problem instances. How to maintain stable learning and reliable performance in **dynamic, streaming environments** is still an open challenge. In particular, single-policy RL agents often become unstable under distribution shift.
>
> * **Discovery:** We show for **the first time** that actor–critic ensembles unlock a practical and powerful solution to the stability–adaptation dilemma in online NCO. RAISE provides a principled blueprint that combines VRAA, decoupled critic ensembles, and decision-aligned updates to make ensemble RL both effective and practical in this setting.
>
> * **Claim:** To the best of our knowledge, RAISE is the first approach to effectively employ actor–critic ensembles for robust online optimization in NCO tasks. This work provides a **strong stepping stone** toward dependable online NCO solvers capable of handling real-world dynamics.
>
> **Revision:**
> We have revised the **Introduction** to explicitly frame the work as filling the "Online Optimization for NCO" gap and highlighting the pioneering role of ensembles in this context:
> > This paper identifies Online Optimization for NCO as an underexplored frontier and presents ensemble-based online RL as a principled solution, paving the way for future progress in this domain.
>
> ### **Q2. Justification for the "Deterministic Execution Time" assumption.**
>
> **Response:**
> Our modeling follows **standard practices** in the **algorithm-focused DWS literature** published in top-tier domain journals [1,2,3,4,5] and AI conferences [6,7,8,9]. The assumption of using deterministic execution times for decision-making is consistent with foundational works in *heuristic design* (e.g., HEFT [1] and GPHH [2]) and *recent RL studies* (e.g., Dense [7], GOODRL [8], GATES [9]).
>
> ### **Q3. Clarification on Queueing Mechanism and Execution Time.**
>
> **Response:**
> We clarify the mechanism in our simulator in **Appendix B** to correct the misunderstanding that queuing delay is ignored.
> * **FIFO Queue Logic:** When a task is assigned to a specific machine $M_q$, it is placed in $M_q$'s **pending queue** and processed strictly in First-In-First-Out (FIFO) order. The machine cannot process a new task until all preceding tasks in its queue are completed.
> * **Strict Separation of Times:** We explicitly distinguish between two time components:
>     * **Execution Time ($et_{ij}$):** This refers *only* to the actual processing duration on the machine (i.e., *TaskFinishTime - TaskStartTime*), which is modeled as deterministic based on workload ($tw/ms$).
>     * **Waiting Time (Queuing Delay):** This is the duration a task waits in the pending queue (i.e., *TaskStartTime - TaskReadyTime*).
>
> **Revision:**
> We have added a definition of waiting time $wt_{ij}^{(q)}$ in **Appendix B** for clarification. All other definitions in Appendix B remain the same, as they do not influence the final objective calculation.
>
> ### **Q4. Clarification on "Queueing Delay" modeling.**
>
> **Response:**
> We clarify that **Queueing Delay is mathematically included** in our Flowtime ($F_i$) calculation.
> * **Definition:** The simulator calculates $F_i = ft_i - at_i$, where $ft_i$ means the finish time of workflow $W_i$ (see Eq.-(13)) and $at_i$ means the arrival time of workflow $W_i$.
> * **Implicit Inclusion:** A workflow consists of a set of tasks with Directed Acyclic Graph (DAG) dependencies. Accordingly, $ft_i$ represents the completion time of the latest finishing task in workflow $W_i$, while $at_i$ represents the workflow's arrival time (i.e., the earliest ready time of its entry tasks).
> * **Conclusion:** The calculated flowtime ($F_i = ft_i - at_i$) inherently encapsulates all intermediate queuing delays and resource contention waiting times along the critical path. This formulation aligns with the standard modeling conventions widely used in the DWS literature (e.g., [1],[6],[7]). Therefore, *queuing delay is not ignored.*

---

> ### Author Response · Authors · 2025-11-27
> **Response to Reviewer KLDs - 2**
>
> ### **Q5. Impact of Decision Latency on Flowtime ($F_i$).**
>
> **Response:**
> The inference latency is millisecond-level, which is negligible in practice.
> * **Hardware:** The reported latency ($\approx 0.06s$ in Table 1) is measured on a **CPU (AMD EPYC 7702 64-Core Processor)**. In an environment equipped with GPUs (e.g., **NVIDIA A100-PCIE-40GB**), our models show inference time drops by $\sim 4\times$ to **0.015s**.
> * **Energy Cost:** The `AMD EPYC 7702 64-Core Processor` has a default TDP (Thermal Design Power) of **200W** [1]. The energy per decision is:
>     $$E_{cpu} \approx 0.0341s \text{ to } 0.0864s \times 200W \approx \mathbf{6.82J - 17.28J}$$
> The TDP of `NVIDIA A100-PCIE-40GB` is **250W** [2], this reduces inference time to $\sim0.0085s - 0.0216s$, yielding:
>     $$E_{gpu} \approx 0.0085s \text{ to } 0.0216s \times 250W \approx \mathbf{2.13J - 5.40J}$$
> These values are negligible compared to the total energy consumed by the cloud resources executing the workflows.
> * **Impact:** Compared to workflow task durations, a **0.015s** delay represents **less than 0.1%** of the total flowtime. Thus, excluding it from $F_i$ does not alter the comparative conclusions.
>
> **Revision:**
> We have added a **Latency Analysis** in **Appendix L**, detailing the energy consumption of using CPU vs. GPU.
>
> ### **Q6. Significance of Performance Gains.**
>
> **Response:**
> We respectfully emphasize that, in large-scale cloud scheduling environments, even a **4\% reduction** in mean flowtime represents a highly meaningful improvement.
> * **Algorithmic Significance:**
>   The widely used HEFT heuristic [1], with over 4600 citations, has been refined for more than two decades. Achieving up to **29.61 percent improvement** over such a mature baseline indicates a substantial advance in scheduling effectiveness rather than an incremental gain.
>
> * **Operational Scale:**
>   Modern hyperscale cloud providers such as AWS, Azure, and Google Cloud manage enormous workflow volumes. For instance, a large data center may process **approximately 500,000 workflows per day**. A 4\% reduction in average flowtime translates to **millions of seconds of end-to-end delay saved daily**, significantly improving user experience and system responsiveness at scale.
>
> * **Economic and Environmental Impact:**
>   Workflow efficiency directly affects the "active time" of high-power servers. Reducing flowtime by a few percent at cloud scale can decrease energy consumption by **thousands of kWh per day**, yielding substantial long-term savings and a measurable reduction in carbon footprint. These improvements align with industry priorities in sustainability and cost-efficient operation.
>
> **Revision:**
> We have added a **Practical Implications** discussion in **Appendix O**.
>
> ---
> **References**
> [1] Topcuoglu et al. "Performance-effective and low-complexity task scheduling for heterogeneous computing." IEEE Transactions on Parallel and Distributed Systems, 13.3 (2002): 260-274.
> [2] Xu et al. "Genetic programming for dynamic workflow scheduling in fog computing." IEEE Transactions on Services Computing, 16.4 (2023): 2657-2671.
> [3] Dong et al. "Deep reinforcement learning for fault-tolerant workflow scheduling in cloud environment." Applied Intelligence, 53.9 (2023): 9916-9932.
> [4] Yu et al. "Integrating cognition cost with reliability QoS for dynamic workflow scheduling using reinforcement learning." IEEE Transactions on Services Computing, 16.4 (2023): 2713-2726.
> [5] Zhu et al. "Learning to optimize workflow scheduling for an edge–cloud computing environment." IEEE Transactions on Cloud Computing, 12.3 (2024): 897-912.
> [6] Jeon et al., "Neural DAG Scheduling via One-Shot Priority Sampling." ICLR 2023.
> [7] Qi et al., "Reinforcement learning for one-shot DAG scheduling with comparability identification and dense reward." NeurIPS2025.
> [8] Yang et al. "Graph assisted offline-online deep reinforcement learning for dynamic workflow scheduling." ICLR 2025.
> [9] Shen et al. "GATES: Cost-aware dynamic workflow scheduling via graph attention networks and evolution strategy." IJCAI 2025.
> [10] https://www.amd.com/en/products/processors/server/epyc/7002-series.html
> [11] https://www.nvidia.com/content/dam/en-zz/Solutions/Data-Center/a100/pdf/A100-PCIE-Prduct-Brief.pdf

---

### Official Review · Reviewer_7eFH · 2025-10-31

**Soundness:** 3
**Presentation:** 2
**Contribution:** 2
**Rating:** 4
**Confidence:** 4

**Summary:**

This paper introduces an ensemble-based reinforcement learning framework that adaptively tunes policies to accommodate non-stationary and evolving environments for dynamic workflow scheduling. It resolves tie situations among different actors using value rankings derived from their corresponding critics, balances adaptation speed and training stability through dual critic networks, and enhances sample efficiency and preserves policy diversity by selectively updating actors.

**Strengths:**

The paper proposes a simple value ranking mechanism to effectively solve the tie problems while ensuring the inference speed.

Instead of updating all actors, the paper updates only those actors responsible for the selected actions, effectively accelerating training and reducing computational overhead while preserving strong policy adaptability. This is a simple yet effective solution for ensemble-based RL methods.

**Weaknesses:**

In the Introduction, you mentioned that most RL-based DWS approaches rely on offline training. This statement is not very accurate and may mislead readers. In fact, classic RL is an online training paradigm by consistently interacting with environments to collect samples and learning policy. I think what you meant is that most current RL methods focus on training the policy from scratch, while neglecting subsequent online tuning of the policy to adapt to changing environments. So please state your point more accurately.

In Section 4.2, you mentioned that adaptive critics may offer accurate estimations but are less stable, and conservative critics provide stability but tend to underestimate action values. However, there seem to be no experimental results supporting this assumption.

In Section 5.4, the experimental results can only demonstrate the adaptability introduced by the decision-aligned sample assignment and gradient control mechanism, but do not show their ability to ensure training stability and enhance sample efficiency claimed in the Abstract.

**Questions:**

There are several recent advances in online tuning within reinforcement learning, often referred to as continual RL. It is recommended that the authors conduct a literature review and select several representative algorithms for comparison.

---

> ### Author Response · Authors · 2025-11-27
> **Response to Reviewer 7eFH - 1**
>
> We thank the reviewer for their constructive and insightful feedback. We have revised the manuscript to improve clarity, strengthen empirical support, and better position our contributions.
>
> ### **Q1. Clarification on the Statement Regarding Offline vs. Online RL in Introduction.**
>
> **Response:**
> Our original intention was to highlight that, in the DWS literature, most DRL-based approaches rely on fixed policies trained offline, with no further tuning.
>
> **Revision:**
> We have revised the statement in **Section 1 (Introduction)** as
> > "However, most existing approaches focus on training policies from scratch, while often neglecting the need for subsequent **online adaptation** to evolving environments."
>
> ### **Q2. Experimental Support for the Distinction of Adaptive vs. Conservative Critics.**
>
> **Response:**
> We have added empirical evidence (detailed in Appendix M) to validate the assumption that *Adaptive Critics track dynamic trends (higher variance) while Conservative Critics provide stable estimates.* We visualized the Q-value distributions of both critic ensembles on the same set of states and actions (see heatmap in Figure 2). We also list the data from the heatmap in **Table R4** for your convenience.
> |**Critics**|**Mean**|**Std.**|**Range**|
> |--|--|--|--|
> |Conservative Critics|-0.8367|0.2821|[-1.7795,-0.0683]|
> |Adaptive Critics|-3.3997|1.6314|[-6.7356,-0.9814]|
>
> **Table R4: Q-value distributions on the same set of states and actions**
>
> The statistical analysis of the heatmap confirms their distinct behaviors:
> * **Stability (Standard Deviation):** The *Conservative Critics* exhibit high stability with a low standard deviation of **0.2821**. In contrast, the *Adaptive Critics* show a much higher standard deviation of **1.6314**. This $\sim 5.78\times$ difference indicates that Adaptive critics are highly sensitive to recent fluctuations, while Conservative critics act as stable anchors.
> * **Accuracy (Value Range):** The *Adaptive Critics* cover a significantly wider value range ([-6.7356, -0.9814]) compared to the *Conservative Critics* ([-1.7795, -0.0683]). This large discrepancy indicates that Conservative estimates, being anchored to prior knowledge, deviate significantly from the current online reality captured by Adaptive critics.
>
> **Revision:**
> We have added a Heatmap in **Section 4.2** and the corresponding analysis to **Appendix M** to empirically substantiate the distinct roles of Adaptive vs. Conservative Critics. We also polished the statement in **Section 4.2**:
> >"...conservative critics provide stability but tend to yield biased action value estimates due to distribution shifts. To validate these distinct behaviors, we visualize the value distributions of both ensembles during online adaptation in Figure 2."
>
> ### **Q3. Evidence for "Training Stability" and "Sample Efficiency" in Section 5.4.**
>
> **Response:**
> We clarify how the ablation results in **Table 3** and a new table in **Appendix N** demonstrates these properties:
> * **Stability:** The `RAISE w/o Grad` variant (which removes gradient control) degrades performance significantly (e.g., objective increases from $390.1$ to $557.45$ in scenario $\langle 5\times7, 9.0, 20k \rangle$). This drastic drop indicates that without our gradient-controlled policy updates, the training process becomes unstable.
> * **Sample Efficiency:** In Table R5, we track the *relative performance improvement* (see metrics in **Section 5.1**) of RAISE and GOODRL compared to the heuristic baseline (HEFT) at the same interaction steps (number of samples). These empirical findings confirm that RAISE improves sample efficiency compared to baselines.
>     * RAISE achieves immediate positive gains. With 147k samples, RAISE already outperforms HEFT by 7.99%, whereas GOODRL performs worse than the heuristic (-2.41%) at the same stage. This indicates RAISE requires significantly less "warm-up" data to learn effective policies.
>     * As training progresses to 345k samples, RAISE’s improvement reaches 23.42%, maintaining a $\sim 2\times$ performance lead over GOODRL (11.74%).
>
>     | |**147k**|**187k**|**226k**|**266k**|**305k**|**345k**|
>     |--|--|--|--|--|--|--|
>     |GOODRL|-2.41%|-0.16%|4.44%|7.79%|10.25%|11.74%|
>     |RAISE|7.99%|12.58%|16.99%|20.18%|22.26%|23.42%|
>
>     **Table R5: Relative performance improvement of baselines over HEFT across online interaction steps**
>
> **Revision:**
> We have polished the stability statement in **Section 5.4**, and added the corresponding analysis of sample efficiency in **Appendix N** to quantitatively support the claim.

---

> ### Author Response · Authors · 2025-11-27
> **Response to Reviewer 7eFH - 2**
>
> ### **Q4. Comparison with Recent Advances in Continual RL.**
>
> **Response:**
> We have expanded **Section 2 (Related Work)** to discuss these advances. Below, we clarify why existing continual RL methods do not directly address the challenges of Online DWS and therefore require substantial adaptation before they can be meaningfully applied.
>
> **1. Conceptual Distinction**
> * **Continual RL** (CRL) typically focuses on learning across multiple distinct tasks, where the objective is to mitigate forgetting and facilitate transfer across known task boundaries [1,2,3,4].
> * **Online RL** in contrast, considers **a single lifelong task** with a continuously evolving data distribution  [5,6].
> * In our Dynamic Workflow Scheduling (DWS) setting, the environment presents **a continuous, unified decision-making process** driven by an evolving workload stream rather than a sequence of discrete tasks. This aligns naturally with **non-stationary Online RL**, rather than existing CRL methods.
>
> This distinction indicates that CRL algorithms target a different problem structure than the one posed by Online DWS.
>
> **2. Challenges in Directly Applying CRL Algorithms**
> While CRL techniques such as plasticity resets [3,4] or loss-of-plasticity strategies from Dohare et al., Nature 2024 [1] are effective in their intended domains, they introduce difficulties when applied to graph neural network (GNN) actor–critic models used in DWS.
>
> * **Destruction of Feature Extraction:** GNNs rely on carefully trained message-passing weights to encode workflow DAG dependencies. Resetting or perturbing these weights can disrupt their ability to capture structural information.
>
> * **Empirical Failure:** Our attempt to adapt the reset-based mechanism from [1] to a GNN actor–critic architecture resulted in unstable value estimates and prevented policy learning from converging, due to disrupted structural representations.
>
> **3. Domain Gap**
> CRL methods do not specify how reset-based mechanisms should be applied to NCO or scheduling tasks (e.g., which layers to reset, how frequently, whether to reset actor and critic jointly), making direct application non-trivial.
>
> **4. RAISE's Position**
> To the best of our knowledge, RAISE is among the first methods to address **Online Reinforcement Learning for Neural Combinatorial Optimization (NCO)** in dynamic, streaming environments. Prior NCO research has focused largely on offline or episodic settings, whereas RAISE is designed for continuous, evolving environments where the agent must adapt to streaming data without sacrificing stability. RAISE therefore opens a new direction for building online NCO solvers that can operate reliably under persistent distribution shifts and long-horizon decision dependencies.
>
> **Revision:**
> We have expanded **Section 2** to include a literature review of recent CRL works (including [1]-[6]):
> > **Continual and Online RL.** Recent advances in continual RL (CRL) tackle stability–plasticity trade-offs using regularization [1], prompting [2], or reset-based strategies that preserve plasticity [3,4]. In online RL, efficient fine-tuning [5] and context-aware adaptation [6] have shown promise. However, these approaches are largely designed for continuous control or multi-task settings, where small errors are recoverable. In contrast, combinatorial scheduling is highly unforgiving: a single suboptimal assignment can propagate through DAG dependencies and trigger system-wide delays. As a result, existing continual or online RL methods cannot be applied directly and require substantial redesign to function in this setting.
> > RAISE fills this gap with a novel ensemble architecture that maintains stability while adapting to non-stationary workload dynamics. To the best of our knowledge, it is among the first methods to bring principled Online RL to the NCO domain.
>
> ---
> **References**
>
> [1] Kirkpatrick et al. "Overcoming Catastrophic Forgetting in Neural Networks." Proceedings of the National Academy of Sciences 114.13 (2017): 3521-3526.
> [2] Wang et al. "Learning to Prompt for Continual Learning." CVPR 2022.
> [3] Dohare et al. "Loss of Plasticity in Deep Continual Learning." Nature 632.8026 (2024): 768-774.
> [4] Farias, Vivek, and Adam Daniel Jozefiak. "Self-Normalized Resets for Plasticity in Continual Learning." ICLR 2025.
> [5] Zhou et al. "Efficient Online Reinforcement Learning Fine-Tuning Need Not Retain Offline Data." ICLR 2025.
> [6] Hamadanian et al. "Online Reinforcement Learning in Non-Stationary Context-Driven Environments." ICLR 2025.

---

### Official Review · Reviewer_rcSX · 2025-10-31

**Soundness:** 3
**Presentation:** 3
**Contribution:** 2
**Rating:** 4
**Confidence:** 4

**Summary:**

This paper proposes RAISE (Robust Actor-Critic Integration for Scheduling Ensembles), which is an online, actor-critic reinforcement learning paradigm for dynamic workflow scheduling.  Taking significant influence from the recent GOODRL (Yang et al., 2025) approach to dynamic workflow scheduling, RAISE extends this approach to ensembles of actors and critics, introducing several improvements.

**Strengths:**

- Demonstrates state-of-the-art performance in minimizing mean flowtime for dynamic workload scheduling, improving upon state-of-the-art performance by up to 5-6% and demonstrating improved robustness of performance across varying workloads and machine configurations.

- Utilizes realistic, highly dynamic workload scheduling scenarios for testing making experimental results highly relevant to real-world scheduling.

- Introduces multiple mechanisms to stabilize system performance, including (i) selective actor updates, to provide direct, relevant gradients to actors, encouraging ensemble diversity and, consequently, system stability, and (ii) dual critic ensembles with decoupled updates, creating critic ensembles with differing update rates, allowing for more balanced value estimation signals and improving overall system stability.

**Weaknesses:**

- The motivation for development of the proposed approach, which significantly overlaps with the offline-online RL approach proposed in Yang et al. (2025), is the unsupported claim that the reliance of the approach in Yang et al. (2025) on a single policy/actor may render it incompetent across a long time period.
- Tie breaking mechanism in value-ranked action aggregation randomly utilizes either the set of Adaptive Critics or the set of Conservative Critics to generate action rankings, rather than utilizing both sets of critics to ensure highest accuracy in all tie-breaking action rankings.  Authors state that this prevents the final action selection from consistently favoring one critic set over the other but do not explain why this is unfavorable, do not explore potential alternative approaches that do utilize both sets of critics, and neglect to include any such alternative approach in their ablation studies (however, see question below).
- Computational overhead of proposed approach is multiple times that of current state-of-the-art approaches, including GOODRL (about 2-3x higher) and GPHH (about 10x higher).  Authors discount this increased computational overhead as practically negligible due to latency in other aspects of one of the real-world environments in which this system would be employed (i.e., cloud computing) but do not address other aspects of computational overhead (e.g., energy consumption) or net impact in other real-world contexts.

**Questions:**

1.	What is the basis for your claim that the single policy/actor approach in Yang et al. (2025) may render it incompetent across a long time period?
2.	With respect to the random selection of the set of critics used in a tie-break valuation, did authors ever consider or investigate any correlation between the critic set utilized and the actor(s) whose action was selected?  That is, did the authors confirm that a given set of critics selected sufficiently evenly across actors across samples to allow for both the adaptive and conservative ensembles to affect the gradients of all actors (i.e., avoiding the potentially unintended creation of adaptive actors and conservative actors)?
3.	Regarding the ablation study of action aggregation methods, did the ensemble with value strategy utilize the dual sets of critic ensembles (adaptive and conservative) and average Q-values across them, or did it utilize only a single type of critic ensemble?

---

> ### Author Response · Authors · 2025-11-27
> **Response to Reviewer rcSX - 1**
>
> We thank the reviewer for the constructive feedback. We have addressed the concerns regarding the motivation of single-policy instability, performed new studies on critic selection, and clarified the overhead analysis. The manuscript will be updated accordingly.
>
> ### **Q1. Motivation Regarding the Long-term Instability of Single-Policy Methods.**
>
> **Response:**
> We clarify the distinct roles of GOODRL and RAISE.
> *  **Scope:** GOODRL primarily targets the **offline-to-online transfer** setting. In this setting, the policy is trained offline under a fixed arrival rate distribution and is then deployed online in environments that initially match this distribution but subsequently experience time-varying arrival rates (increasing or decreasing). This requires the policy to generalize beyond its stationary offline training regime and adapt to the evolving workload dynamics.
> In comparison, RAISE addresses **long-term non-stationarity** in highly volatile environments (changing arrival rate), as shown in Figure~2 in the manuscript. Online DWS scenarios often exhibit substantial fluctuations in arrival rates, creating distributional shifts that can be difficult for a single policy to track reliably over time. In our experiments, such shifts lead to noticeable degradation in single-policy baselines, whereas RAISE maintains stable performance through ensemble-based adaptation.
>
> *  **Error Propagation:** In online DWS, decision-making is sequential and irreversible. If a single actor overfits to recent noise and makes an aggressive, suboptimal assignment on a critical bottleneck task, this error propagates, delaying all dependent successor tasks. Single-policy agents lack mechanisms to cope with such changes.
> *  **Ensemble Stability:** RAISE leverages "collective wisdom" to mitigate this risk. The ensemble filters out "radical" behaviors, ensuring that one actor’s estimation error does not compromise the upcoming schedules. However, actors frequently disagree (16%-23%) as shown in our tie-breaking analysis (Figure~4 in Appendix A), requiring new mechanisms to achieve effective online learning.
> * **Novelty Claim:** To the best of our knowledge, RAISE is **the first approach** to effectively employ actor–critic ensembles for robust online optimization in NCO tasks. This work provides a **strong stepping stone** toward dependable online NCO solvers capable of handling real-world dynamics.
>
> **Revision:**
> We have revised **Section 1 (Introduction)** to explicitly contrast GOODRL’s focus on initial adaptation with **RAISE’s emphasis on long-term robustness** against error propagation.
>
>
> ### **Q2. Justification for "Random Selection" vs. "Using Both" Critics.**
>
> **Response:**
> We appreciate the reviewer's perspective. However, our additional analysis (Appendix J) indicates that the assumption that using both critic sets simultaneously yields the highest accuracy does not consistently hold. In contrast, the random strategy preserves clearer distinctions between adaptive and conservative evaluations and performs more reliably in practice. Concretely, we compared the relative improvement of "Random" and "Both" over single-policy methods (e.g., GOODRL). As shown in Table R1 below (detailed in Appendix J), "Both" does not outperform "Random" while noticeably increasing the inference cost. Using "Both" (e.g., averaging ranks) dilutes the distinct **adaptive** vs. **conservative** signals.
>
> | **Mechanism**       | **60-th** | **80-th** | **100-th** | **120-th** | **140-th** | **160-th** |
> |---------------------|-----------|-----------|------------|------------|------------|------------|
> | Random              | 10.16%    | 12.73%    | 13.13%     | 13.43%     | 13.39%     | 13.23%     |
> | Both                | 10.08%    | 11.81%    | 12.06%     | 12.11%     | 12.49%     | 12.24%     |
>
> **Table R1: Relative improvement over single-policy (GOODRL) in cumulative flowtime/workload**
>
> **Revision:**
> We have added these results to **Appendix J**, and claimed the metric of relative Improvement in **Section 5.1**.
>
>
> ### **Q3. Clarification on the Critics used in the "Ensemble with value" Ablation.**
>
> **Response:**
> To ensure a fair comparison of the **aggregation mechanism itself** (Value vs. Rank), we used the *Conservative Critic Ensemble* for both the "Ensemble with value" baseline and "Ensemble with VRAA" in Table 2 of the manuscript. The goal was to isolate the impact of the aggregation logic, while using different critics would introduce confounding variables. Our results show that aggregating by *Rank* is robust to scale differences.
>
> **Revision:**
> We have updated the corresponding text in **Section 5.3** to explicitly state:
> > "To isolate the effectiveness of the aggregation logic, both methods utilize the same set of pre-trained Conservative Critics."

---

> ### Author Response · Authors · 2025-11-27
> **Response to Reviewer rcSX - 2**
>
> ### **Q4. Are Actors Selected by Critics Sufficiently Evenly?**
>
> **Response:**
> We track the selection probability of 5 actors under different critic sets in Table R2 (see Appendix K for details), and have the following observations.
> * **No Extreme Polarization:** We did not find any specific actor being significantly ignored by any given set of critics. When using *only Adaptive* or *only Conservative* critics, actor selection probabilities remained balanced. While different critics (*Adaptive vs. Conservative*) adjust the probabilities, every actor continues to receive gradients.
> * **Drivers of Uniformity:** Our results suggest that the degree of uniformity in action selection is influenced more by performance differences among actors than by the choice of critic type.
>     * **High Variance Actor Group:** In Table R2, where actors have larger performance gaps, selection probabilities ranged from 15.93% to 22.81%.
>     * **Low Variance Actor Group:** In Table R3, where actors have close performance, selection probabilities were more uniform, ranging from 18.26% to 21.69%.
>
>         | **Type**   | **Env steps** | **Actor-1** | **Actor-2** | **Actor-3** | **Actor-4** | **Actor-5** | **Sum** |
>         |--|--|--|--|--|--|--|--|
>         | **Adpative**   | 10k           | 20.59%      | 16.21%      | 20.66%      | 20.10%      | 22.44%      | 100%    |
>         | | 30k           | 17.68%      | 18.34%      | 20.99%      | 20.53%      | 22.46%      | 100%    |
>         | | 50k           | 17.15%      | 18.83%      | 20.91%      | 20.59%      | 22.52%      | 100%    |
>         | | 70k           | 16.83%      | 19.33%      | 21.13%      | 19.90%      | **22.81%**      | 100%    |
>         | **Conservative**| 10k           | 20.52%      | **15.93%**      | 20.97%      | 20.13%      | 22.45%      | 100%    |
>         | | 30k           | 17.68%      | 17.79%      | 21.31%      | 20.65%      | 22.57%      | 100%    |
>         | | 50k           | 17.37%      | 18.10%      | 21.06%      | 20.96%      | 22.51%      | 100%    |
>         | | 70k           | 17.27%      | 18.49%      | 21.13%      | 20.41%      | 22.70%      | 100%    |
>
>         **Table R2: Actor selection statistics on a high variance group of actors**
>         | **Type**   | **Env steps** | **Actor-1** | **Actor-2** | **Actor-3** | **Actor-4** | **Actor-5** | **Sum** |
>         |--|--|--|--|--|--|--|--|
>         | **Adpative**   | 10k           | 20.36%      | 21.33%      | 19.21%      | **18.26%**      | 20.84%      | 100%    |
>         | | 30k           | 20.73%      | **21.69%**      | 19.87%      | 18.52%      | 19.18%      | 100%    |
>         | | 50k  | 20.99%      | 20.78%      | 20.55%      | 19.31%      | 18.37%      | 100%    |
>         | | 70k | 21.26%      | 19.87%      | 20.94%      | 19.59%      | 18.35%      | 100%    |
>         | **Conservative**| 10k           | 20.32%      | 20.93%      | 19.63%      | 18.57%      | 20.54%      | 100%    |
>         | | 30k | 20.23%      | 21.43%      | 20.23%      | 18.95%      | 19.16%      | 100%    |
>         |  | 50k | 20.45%      | 20.62%      | 20.76%      | 19.72%      | 18.44%      | 100%    |
>         | | 70k | 20.73%      | 19.86%      | 21.04%      | 19.88%      | 18.49%      | 100%    |
>
>         **Table R3: Actor selection statistics on a low variance group of actors**
>
> **Revision:**
> We have added the actor selection statistics analysis in **Appendix K**.
>
> ### **Q5. Real-world Overhead and Energy Consumption.**
>
> **Response:**
> We have expanded the overhead analysis to include a direct energy consumption calculation using the hardware specifications.
> * **CPU Energy:** The **AMD EPYC 7702 64-Core Processor** has a default TDP (Thermal Design Power) of **200W** [1]. Based on the inference times in Table 1 ($0.0341s - 0.0864s$), the energy per decision is:
>     $$E_{cpu} \approx 0.0341s \text{ to } 0.0864s \times 200W \approx \mathbf{6.82J - 17.28J}$$
> * **GPU Energy:** If deployed on an **NVIDIA A100-PCIE-40GB** (TDP **250W** [2]), RAISE inference typically achieves a speedup of $\sim4\times$ compared to CPUs. This reduces inference time to $\sim0.0085s - 0.0216s$, yielding:
>     $$E_{gpu} \approx 0.0085s \text{ to } 0.0216s \times 250W \approx \mathbf{2.13J - 5.40J}$$
>
> These values are negligible compared to the total energy consumed by the cloud resources executing the workflows. A single workflow task running for just 1 minute on a similar machine consumes $\approx 12,000J$ ($60s \times 200W$). Thus, the energy cost of RAISE's decision-making is **orders of magnitude lower (<0.1%)** than the execution energy it manages.
>
> **Revision:**
> We have added an *Energy Efficiency Analysis* in **Appendix L**, detailing these calculations to demonstrate that the computational overhead is economically and environmentally insignificant.
>
> ---
> [1] https://www.amd.com/en/products/processors/server/epyc/7002-series.html
> [2] https://www.nvidia.com/content/dam/en-zz/Solutions/Data-Center/a100/pdf/A100-PCIE-Prduct-Brief.pdf

---

### Author Response · Authors · 2025-11-27
**Summary of Revisions**

Dear Reviewers,

We sincerely thank all the reviewers for their constructive feedback. Your insights have been invaluable in strengthening our work.

In the revised manuscript (changes highlighted in **red**), we have significantly strengthened the work by:
* **Clarifying Scope & Novelty:** We explicitly position RAISE as *one of the first works dedicated to principled Online RL for Neural Combinatorial Optimization (NCO)*, an underexplored frontier. To the best of our knowledge, it is **the first approach** to effectively leverage actor-critic ensembles to achieve stable online adaptation, providing a strong stepping stone toward dependable real-world solvers.
* **Validating Mechanisms:** Providing rigorous empirical backing for our design choices, including **heatmap analyses** of Dual Critics, **frequency statistics** for Gradient Control, and **ablation studies** justifying the VRAA tie-breaking strategy.
* **Demonstrating Impact:** Adding new analyses on **Sample Efficiency** (\~23% gain over heuristics), **Scalability** (n=5 trade-off), and **Energy Efficiency** (inference cost <0.1\%).

We believe these revisions comprehensively address the concerns regarding novelty, justification, and significance. We would greatly appreciate your consideration of adjusting your rating to reflect the revisions.

Best regards,
Authors of Paper 3965

---

### Meta-Review · Area_Chair_BsXu · 2025-12-16

**Summary:**

This paper targets a practical problem on dynamic workflow scheduling under non-stationary workloads. It proposes an ensemble-based online RL approach named RAISE, which demonstrates strong through extensive empirical evaluations.

- Some reviewers viewed the proposed method RAISE as an incremental extension and requires the authors to clearly separate it in terms of  what is new, e.g., value-ranked tie-breaking, dual-critic roles, and stability mechanisms, and why single-policy methods fail under distribution shift. In the rebuttal, the authors explain that the failure mode of single-policy approaches under non-stationary shifts and position RAISE as maintaining decision robustness via ensemble diversity and actor selection and aggregation.
- Some reviewers asked for justification of key components including but not limited to why tie-break with value-ranked actors, why random vs "both" critics, what "ensemble with value" ablation truly measures, etc.
- Reviewers also questions the assumptions and realism in the simulator and modeling. For example, whether the assumptions like near-deterministic machine execution time and treatment of queueing delay reflect real systems and whether the study captures major practical factors. Those gaps need to be justified potentially from the perspectively of both theoretically/algorithmically and empirically.
- Another concern on the practicality of the proposed RAISE is on its compute overhead and scalability. Multiple comments asked whether ensemble inference and training overhead is negligible in practice, how performance scales with ensemble size, and whether complexity outweighs gains.

**Reviewer Concerns:**

As mentioned above, some concerns have been addressed during the rebuttals such as the authors explain that the failure mode of single-policy approaches under non-stationary shifts and position RAISE as maintaining decision robustness via ensemble diversity and actor selection and aggregation. In addition, the rebuttal provides an empirical comparison and explains that mixing adaptive and conservative signals can blur distinct roles, and the authors add explicit estimates of decision latency, energy cost to argue inference overhead is negligible relative to workload execution.

There are some other key concerns that are outstanding or partially addressed:
- One key concern is still on the assumptions and the practicality of the proposed framework. Although the queueing-delay misunderstanding is resolved, skepticism remains about whether the simulator assumptions, e.g., determinism and what factors are excluded, are sufficiently realistic to warrant strong claims about cloud-scale deployment.
- The authors' rebuttal reports that the overhead of the proposed RAISE is small and also provides scaling vs ensemble size analysis. However, this does not directly address the reviewer's question or concern on the operational complexity such as tuning, deployment constraints, sensitivity under abrupt shifts.

**Reviewer Scores:**

Reviewer rcSX raised questions on justification and clarity, which somehow were addressed by the rebuttal.

Reviewer 7eFH's main concerns were the "offline vs. online" phrasing and positioning, as well as missing comparisons to recent continual RL literature. The authors' rebuttal corrects the statement and provides a structured argument for why the setting differs, plus additional sample-efficiency evidence. The positioning to recent works are still weak and needs to be fully incorporated into the revision.

Reviewer KLDs raises deeper realism and modeling issues such as on the assumptions, determinism, and whether delays are properly captured. The rebuttal clarifies queue-delay modeling and argues assumptions are standard in algorithm-focused DWS literature. However, some of those statements are not quite convincing and the reviewer may remain unconvinced without stronger real-system validation.

Reviewer wFb3 was positive on the method and experiments, with remaining questions focusing on complexity and specific design choices. After the rebuttal, this reviewer may remain positive.

---

### Decision · Program_Chairs · 2026-01-26

Reject